

# Learning the ground state of a non-stoquastic quantum Hamiltonian in a rugged neural network landscape

**Marin Bukov[1,2⋆], Markus Schmitt[1] and Maxime Dupont[1,3]**

**1** Department of Physics, University of California at Berkeley, Berkeley, USA
**2** Department of Physics, St. Kliment Ohridski University of Sofia, Sofia, Bulgaria
**3** Materials Sciences Division, Lawrence Berkeley National Laboratory, Berkeley, USA

⋆ mgbukov@phys.uni-sofia.bg

## Abstract

Strongly interacting quantum systems described by non-stoquastic Hamiltonians exhibit rich low-temperature physics. Yet, their study poses a formidable challenge, even for state-of-the-art numerical techniques. Here, we investigate systematically the performance of a class of universal variational wave-functions based on artificial neural networks, by considering the frustrated spin-1/2 $J_1 - J_2$ Heisenberg model on the square lattice. Focusing on neural network architectures without physics-informed input, we argue in favor of using an ansatz consisting of two decoupled real-valued networks, one for the amplitude and the other for the phase of the variational wavefunction. By introducing concrete mitigation strategies against inherent numerical instabilities in the stochastic reconfiguration algorithm we obtain a variational energy comparable to that reported recently with neural networks that incorporate knowledge about the physical system. Through a detailed analysis of the individual components of the algorithm, we conclude that the rugged nature of the energy landscape constitutes the major obstacle in finding a satisfactory approximation to the ground state wavefunction, and prevents learning the correct sign structure. In particular, we show that in the present setup the neural network expressivity and Monte Carlo sampling are not primary limiting factors.



# 1 Introduction

Understanding the effect of many-body interactions in quantum systems is a long-standing challenge of modern physics: the exponential growth of the Hilbert space with the number of particles makes solving the Schrödinger equation impossible beyond a few dozen degrees of freedom. Over the last decades, the advent of sophisticated computational methods has led to breakthroughs in addressing the quantum many-body problem: quantum Monte Carlo techniques [1], tensor network approaches [2,3] and the dynamical mean-field theory framework [4] are now standard tools when it comes to studying strongly correlated systems. However, each of these methods can only address specific classes of problems efficiently, and many remain yet to be solved.

Prominent among the challenging problems is that of frustrated models [5,6]. In antiferromagnetic (AF) systems, for instance, frustration may prevent conventional Néel order at low temperatures and lead instead to exotic magnetic states such as spin liquids [7–9]. A paradigmatic example of such a system is the AF spin-1/2 $J_1 - J_2$ model on the square lattice. Its Hamiltonian reads as

$$\mathcal{H} = J_1 \sum_{\langle i,j \rangle} \boldsymbol{S}_i \cdot \boldsymbol{S}_j + J_2 \sum_{\langle\langle i,j \rangle\rangle} \boldsymbol{S}_i \cdot \boldsymbol{S}_j , \tag{1}$$

where $\boldsymbol{S}_i = (S_i^x, S_i^y, S_i^z)$ is the spin-1/2 operator acting on lattice site $i$, with a total of $N = L \times L$ lattice sites; we adopt periodic boundary conditions and we assume $J_1, J_2 \geq 0$. The notation $\langle i, j \rangle$ and $\langle\langle i, j \rangle\rangle$ restricts the sum to nearest and next-nearest neighbor spins, respectively. Working in the $S^z$ basis, denoted by $|\boldsymbol{s}\rangle \equiv |\boldsymbol{s}_1, \boldsymbol{s}_2, \ldots, \boldsymbol{s}_N\rangle$ with $\boldsymbol{s}_i = \uparrow, \downarrow$, the normalized ground state of Eq. (1) can be written without loss of generality as,

$$|\Psi_{\mathrm{gs}}\rangle = \sum_{\{\boldsymbol{s}\}} c_{\boldsymbol{s}} |\boldsymbol{s}\rangle, \text{ with } c_{\boldsymbol{s}} \in \mathbb{R} . \tag{2}$$

For $J_1 = 0$ or $J_2 = 0$, the system is frustration-free, and AF interactions display a bipartite pattern with sublattices A and B. Therefore, the Marshall-Peierls sign rule applies [10–12]: the sign of the ground state coefficients $c_{\boldsymbol{s}}$ is fully determined by the parity of the total number of spins pointing up or down on one of the sublattices, i.e., $\mathrm{sign}(c_{\boldsymbol{s}}) = (-1)^{N_{\uparrow \in A}(\boldsymbol{s})}$. This prior knowledge allows one to perform a sublattice spin rotation to make the ground state wavefunction positive, eliminating the infamous sign-problem in quantum Monte Carlo simulations [13,14]. At $J_2 = 0$, the system displays Néel order with wavevector $\boldsymbol{q} = (\pi, \pi)$ [15,16], and collinear stripe order with wavevector $\boldsymbol{q} = (0, \pi)$ or $(\pi, 0)$ for $J_1 = 0$.

Away from the two values $J_1 = 0$ and $J_2 = 0$, frustration sets in, and the Marshall-Peierls sign rule does not hold anymore. The lack of efficient numerical methods makes the phase diagram elusive, with discordant scenarios tracing back to the early days of research on high-temperature cuprate superconductors [17–54]. Although most studies point toward the existence of an intermediate phase (or two) in a small parameter window around $J_2/J_1 \approx 0.5$, sandwiched between the Néel and stripe phases, its existence remains controversial. Proposed ground state candidates go from columnar [21,22,28] or plaquette [29,31,39,45] valence-bond states to gapped [40] or gapless [44,51] spin liquids.

The advent of machine learning (ML) techniques [55–57] has brought new hope in addressing challenging many-body problems, with neural networks being used as a generic variational ansatz [58]. In particular, much like the `MNIST` data set of handwritten digits in the ML community [59], the ground state search of the frustrated $J_1 - J_2$ model has recently turned into a test bed for ideas attempting to push the boundaries of neural-network-based variational approaches [60–65]. Such studies currently receive considerable attention for both equilibrium [66–72] and non-equilibrium [73–79] problems.

A noteworthy difference between typical ML problems and neural quantum states is the high precision that variational quantum many-body simulations aim to achieve to resolve the

exact ground state. In that respect, neural quantum states pose their own interesting technical challenges.

## 1.1 State of the art

Contrary to classifying handwritten digits, finding the ground state of the $J_1 - J_2$ model at $J_2/J_1 \approx 0.5$ still appears to pose a significant challenge, even for state-of-the-art approaches. Recent efforts to employ neural networks for this problem can be summarized in two categories: (i) expanding some otherwise physically motivated variational ansatz, and (ii) pure (i.e., end-to-end) neural network wave functions. Whereas (i) enabled the detailed study of ground state properties even in the most challenging regime around $J_2/J_1 \approx 0.5$ [60, 62, 65, 80], achieving sufficient accuracy with neural network wave functions without physics-informed input remains a formidable challenge despite continuing efforts [61, 63, 64, 70, 80].

Away from the highly frustrated point $J_2/J_1 \approx 0.5$, a deep convolutional neural network (CNN) was shown in Ref. [61] to perform on par with, and even improve upon, existing density matrix renormalization group [40, 45, 50, 81, 82] and standard variational quantum Monte Carlo (VQMC) [44, 46] simulations, while involving fewer variational parameters. It was also demonstrated that one can employ neural networks to enhance the performance of variational Gutzwiller states [62]. Similarly, endowing pair-product states with a neural network, enabled the authors of Ref. [65] to identify two different phases in the vicinity of $J_2/J_1 \approx 0.5$, one of which a spin liquid with fractionalized spinons. Moreover, in Ref. [80], which appeared during the preparation of this manuscript, substantial advancements in accuracy are reported. In that case, the non-stoquasticity of the Hamiltonian is alleviated by incorporating the Marshall-Peierls rule and the gain in accuracy is attributed to the symmetrization of very large Restricted Boltzmann Machines with quantum number projections.

While these studies show that neural networks are sufficiently versatile to find a pretty good approximation to the ground state — and excited states [80, 83, 84] — in a large part of parameter space, the region $J_2/J_1 \approx 0.5$ appears to be an intriguing exception, in particular for pure neural network states without extra physics input. Although learning the Marshall-Peierls rule, which governs the signs in the antiferromagnetic ($J_2 = 0$) and striped ($J_1 = 0$) phases is feasible [64, 70], this has remained elusive on the square lattice for $J_2/J_1 \approx 0.5$. It was recently shown that a generic problem arises using deep-learning-based variational Monte Carlo in frustrated systems: learning the signs of the expansion coefficients in the $S^z$-basis, has increased complexity [63].

At first sight, these difficulties seem to be at odds with the fact that neural networks are universal function approximators in the limit of sufficiently large network size [85–87], and should therefore constitute a suited variational ansatz class that does not require further physical insight.

## 1.2 Overview of the main results

Our goal in this work is to provide a detailed account of the performance of pure neural network wave functions (i.e., without using extra physics-informed input) optimized by VQMC to encode the ground state at the maximally frustrated point $J_2/J_1 = 0.5$. We emphasize that our focus is on understanding the methodological challenges. Therefore, we restrict our discussion to system sizes up to $N = 6 \times 6$ spins, which already exhibit the typical difficulties, and for which we can still obtain reference data using exact diagonalization for comparison.

As a result of a variety of numerical experiments, the main findings presented in the following are:

- Holomorphic networks generalize poorly due to unbounded output and because of the Cauchy-Riemann constraint on gradients. The latter induces potentially restrictive cor-

relations between the phase and amplitude output of the network which limits the learning capabilities of the ansatz. We show that these issues can be mitigated using a non-holomorphic (but still complex-valued) bounded ansatz, which contains two decoupled real-valued networks for the phase and logarithm of the absolute value, of the variational probability amplitude.

- The Marshall-Peierls sign rule appears to be a universal attractor (in the optimization dynamics) even for unbiased networks. The variational parameters which give rise to this behavior define a saddle in the variational energy manifold that is difficult to escape due to the existence of only a few yet high-curvature directions to decrease energy.

- The expressivity of the quantum neural state ansatz is presently not the limiting factor to find the ground state. Monte-Carlo sampling noise does not provide the bottleneck either.

- The rugged bottom of the energy landscape generically renders the optimization difficult, and the obtained results – hard to reproduce. This behavior is caused by an energy landscape topography which features deep valleys, each of which can host many Marshall-Peierls-like saddles. Different optimization runs eventually get trapped into one of these saddles. As a result, increasing the number of parameters in the ansatz is not guaranteed to lead to an improved variational energy at $J_2/J_1 = 0.5$.

## 1.3  Outline

This paper is organized as follows. In Sec. 2, we introduce variational quantum Monte Carlo for the ground state search problem and define the main quantities of interest, such as energy density and energy variance. We also present the optimization algorithm. Then, we introduce deep neural networks as a variational ansatz for quantum many-body spin systems. In Sec. 3, we first introduce two alternative neural network architectures: (i) a single holomorphic network which approximates simultaneously the phase and the amplitude of the wavefunction, and (ii) two decoupled real-valued networks, to approximate the phase and amplitude separately. We provide arguments in favor of using real-valued networks, and exhibit the nature of two subtle numerical instabilities that occur in the ground state search for non-stoquastic Hamiltonians. We also compare the performance of the two architectures. Section 4 is devoted to the problem of learning the correct sign structure of the ground state. We introduce the concept of phase distribution and use it to interpret and analyze training bottlenecks. In pursuit of understanding which part of the algorithm prevents learning the correct phase structure, we perform a number of numerical experiments in Sec. 5, and argue that the expressivity of the ansatz and the Monte Carlo sampling noise do not constitute primary limiting factors. Last, in Sec. 6, we investigate the problem from the perspective of landscape optimization. We draw practical conclusions, which suggest the glassiness of the underlying rugged landscape for the $J_1 - J_2$ model. Finally, we summarize our work, discuss some open problems, and establish connections to other research directions in Sec. 7. A number of details are presented in the appendices for the interested reader, among which a comparison of various optimization algorithms (App. A), a discussion on building physical symmetries into the ansatz (App. C), a stable procedure to initialize the training parameters of deep neural states (App. D), and local versus global Monte-Carlo sampling updates (App. E).

## 2 Variational quantum Monte Carlo with neural network states

### 2.1 Variational quantum Monte Carlo

The variational principle allows one to search for an approximate ground state of a given Hamiltonian $\mathcal{H}$. It is achieved by parametrizing a trial wavefunction with a set of variational parameters $\boldsymbol{\theta}$ in a fixed computational basis $\{\boldsymbol{s}\}$, i.e, $|\psi_{\boldsymbol{\theta}}\rangle = \sum_{\{\boldsymbol{s}\}} \psi_{\boldsymbol{\theta}}(\boldsymbol{s})|\boldsymbol{s}\rangle$. Instead of being parametrized by an exponential number of coefficients as in Eq. (2), the quantum state is parametrized by a chosen function $\psi_{\boldsymbol{\theta}}(\boldsymbol{s})$ with a controllable number of variational parameters.

To find an approximation to the ground state, one minimizes the energy,

$$E_{\boldsymbol{\theta}} = \frac{\langle \psi_{\boldsymbol{\theta}}|\mathcal{H}|\psi_{\boldsymbol{\theta}}\rangle}{\langle \psi_{\boldsymbol{\theta}}|\psi_{\boldsymbol{\theta}}\rangle} = \sum_{\{\boldsymbol{s}\}} p_{\boldsymbol{\theta}}(\boldsymbol{s}) E_{\boldsymbol{\theta}}^{\mathrm{loc}}(\boldsymbol{s}) \geq E_{\mathrm{gs}}, \tag{3}$$

by finding the optimal values for the parameters $\boldsymbol{\theta}$. Here, the exact ground state energy of $\mathcal{H}$ is noted $E_{\mathrm{gs}}$, and

$$E_{\boldsymbol{\theta}}^{\mathrm{loc}}(\boldsymbol{s}) = \frac{1}{\psi_{\boldsymbol{\theta}}(\boldsymbol{s})} \sum_{\{\boldsymbol{s}'\}} \mathcal{H}_{\boldsymbol{s}\boldsymbol{s}'} \psi_{\boldsymbol{\theta}}(\boldsymbol{s}'), \tag{4}$$

is the so-called local energy; it involves the Hamiltonian matrix elements $\mathcal{H}_{\boldsymbol{s}\boldsymbol{s}'} = \langle \boldsymbol{s}|\mathcal{H}|\boldsymbol{s}'\rangle$. For a generic complex-valued variational wavefunction, $E_{\boldsymbol{\theta}}^{\mathrm{loc}}(\boldsymbol{s})$ is also complex-valued. The sum over $\boldsymbol{s}'$ in Eq. (4) is easily performed for local Hamiltonians for which $|\{\boldsymbol{s}'\}| \sim \mathcal{O}(N)$.

Note that

$$p_{\boldsymbol{\theta}}(\boldsymbol{s}) = |\psi_{\boldsymbol{\theta}}(\boldsymbol{s})|^2 \big/ \langle \psi_{\boldsymbol{\theta}}|\psi_{\boldsymbol{\theta}}\rangle \tag{5}$$

defines a proper probability distribution. It enables the use of Monte Carlo (MC) sampling in order to evaluate the energy of Eq. (3) without having to perform the sum "$\sum_{\{\boldsymbol{s}\}}$" over the full Hilbert space H: this procedure becomes infeasible if H is too large, e.g., $\dim(\mathrm{H}) = 2^N$ for the spin-half system considered. Using $N_{\mathrm{MC}}$ Monte Carlo samples, we can approximate the total energy of Eq. (3) by its sample estimate

$$E_{\boldsymbol{\theta}} = \big\langle\!\big\langle E_{\boldsymbol{\theta}}^{\mathrm{loc}} \big\rangle\!\big\rangle \approx \frac{1}{N_{\mathrm{MC}}} \sum_{n=1}^{N_{\mathrm{MC}}} E_{\boldsymbol{\theta}}^{\mathrm{loc}}(\boldsymbol{s}_n), \tag{6}$$

with $\langle\!\langle \cdot \rangle\!\rangle = \sum_{\{\boldsymbol{s}\}} p_{\boldsymbol{\theta}}(\boldsymbol{s})(\cdot)$ denoting the expectation value with respect to the variational probability $p_{\boldsymbol{\theta}}$. The deviation of the MC estimate from the exact quantum expectation value vanishes as $1/\sqrt{N_{\mathrm{MC}}}$ in the limit of $N_{\mathrm{MC}} \to \infty$.

Similarly, the energy variance takes the form,

$$\sigma_{E_{\boldsymbol{\theta}}}^2 = \frac{\langle \psi_{\boldsymbol{\theta}}|\mathcal{H}^2|\psi_{\boldsymbol{\theta}}\rangle}{\langle \psi_{\boldsymbol{\theta}}|\psi_{\boldsymbol{\theta}}\rangle} - \frac{\langle \psi_{\boldsymbol{\theta}}|\mathcal{H}|\psi_{\boldsymbol{\theta}}\rangle^2}{\langle \psi_{\boldsymbol{\theta}}|\psi_{\boldsymbol{\theta}}\rangle} = \big\langle\!\big\langle |E_{\boldsymbol{\theta}}^{\mathrm{loc}}|^2 \big\rangle\!\big\rangle_{\mathrm{c}}, \tag{7}$$

with $|E_{\boldsymbol{\theta}}^{\mathrm{loc}}|^2 \equiv E_{\boldsymbol{\theta}}^{\mathrm{loc}}(E_{\boldsymbol{\theta}}^{\mathrm{loc}})^*$, where $*$ stands for complex conjugation. To simplify the notation, we use $\langle\!\langle AB \rangle\!\rangle_{\mathrm{c}} = \langle\!\langle AB \rangle\!\rangle - \langle\!\langle A \rangle\!\rangle \langle\!\langle B \rangle\!\rangle$ throughout the paper. Note that for exact eigenstates, the energy-variance vanishes. In particular, for the ground state one obtains $E_{\boldsymbol{\theta}}^{\mathrm{loc}}(\boldsymbol{s}) \equiv E_{\mathrm{gs}}$ for all spin configurations $\boldsymbol{s}$ independently.

Now that we have introduced the Monte-Carlo procedure which allows us to estimate the energy for large systems, the next step aims at optimizing the variational parameters. The most straightforward strategy is to directly minimize the energy of the system by seeking an expression for the energy gradient with respect to $\boldsymbol{\theta}$, which can be estimated using Monte Carlo. Labeling each parameter with an index $k$, one arrives at

$$\partial_{\theta_k} E_{\boldsymbol{\theta}} = 2\mathrm{Re}\big(F_k\big), \tag{8}$$

with $\boldsymbol{F}$ the so-called force vector,

$$\boldsymbol{F}_k = \sum_{\{s\}} p_{\boldsymbol{\theta}}(\boldsymbol{s}) \left( \partial_{\boldsymbol{\theta}_k} \ln \psi_{\boldsymbol{\theta}}^*(\boldsymbol{s}) \left[ E_{\boldsymbol{\theta}}^{\text{loc}}(\boldsymbol{s}) - E_{\boldsymbol{\theta}} \right] \right) = \left\langle\!\left\langle O_k^* E_{\boldsymbol{\theta}}^{\text{loc}} \right\rangle\!\right\rangle_{\text{c}}, \tag{9}$$

where $\boldsymbol{O}_k(\boldsymbol{s}) = \partial_{\boldsymbol{\theta}_k} \ln \psi_{\boldsymbol{\theta}}(\boldsymbol{s})$.

Using gradient descent, see App. A, the parameters can be optimized iteratively until convergence of the energy is achieved,

$$\boldsymbol{\theta}_k \longleftarrow \boldsymbol{\theta}_k - 2\gamma \text{Re}\big(\boldsymbol{F}_k\big), \tag{10}$$

with $\gamma \in \mathbb{R}^+$ the step size, a small hyperparameter that one controls the optimization speed in the simulation.

Because the variational manifold may have varying curvature in different directions of parameter space, the Euclidean gradient descent of Eq. (10) can be improved by introducing an appropriate metric [69]. This approach is known as Stochastic Reconfiguration (SR) [88–91], and the optimizations of the parameters take the form,

$$\boldsymbol{\theta}_k \longleftarrow \boldsymbol{\theta}_k - \gamma \sum_{k'} \boldsymbol{S}_{kk'}^{-1} \boldsymbol{F}_{k'}, \tag{11}$$

where the hermitian curvature matrix $\boldsymbol{S}$ is given by,

$$\boldsymbol{S}_{kk'} = \left( \sum_{\{s\}} p_{\boldsymbol{\theta}}(\boldsymbol{s}) O_k^*(\boldsymbol{s}) O_{k'}(\boldsymbol{s}) \right) - \left( \sum_{\{s\}} p_{\boldsymbol{\theta}}(\boldsymbol{s}) O_k^*(\boldsymbol{s}) \right) \left( \sum_{\{s\}} p_{\boldsymbol{\theta}}(\boldsymbol{s}) O_{k'}(\boldsymbol{s}) \right) = \left\langle\!\left\langle O_k^* O_{k'} \right\rangle\!\right\rangle_{\text{c}}. \tag{12}$$

We evaluate the performance of variational quantum Monte Carlo (VQMC) by monitoring the energy per site and the density of energy variance of $|\psi_{\boldsymbol{\theta}}\rangle$. Since our goal is to exhibit the reasons for the highly frustrated $J_2/J_1 = 0.5$ point to defy the VQMC approach, we focus exclusively on small lattices of size $N = 4 \times 4$ and $N = 6 \times 6$, which can be directly compared to exact diagonalization simulations. Moreover, we perform a number of experiments where we evaluate the sums "$\sum_{\{s\}}$" over the entire basis exactly and not stochastically. We refer to these simulations as *full basis* simulations.

## 2.2 Neural network architectures

We now now turn our attention to the variational ansatz itself for the quantum state $\psi_{\boldsymbol{\theta}}(\boldsymbol{s})$. Recently, neural networks emerged as a promising class of variational wave functions [58]. They are universal function approximators in the limit of infinite depth or width (controlling the number of variational parameters): the neural network expressivity theorem asserts that the accuracy of approximating arbitrary functions can be systematically enhanced by increasing the number of network parameters [85–87]. In quantum many-body physics, neural networks have been shown to be capable of encoding volume-law entanglement [92,93], and thus they are believed to represent an alternative to established numerical techniques based on matrix product states. In principle, neural networks are also insensitive to the spatial dimensionality of the physical system of interest. Finally, with the advent of modern automatic differentiation techniques [94], such as the backpropagation algorithm [55,95,96], neural networks are amenable to efficient larger-scale gradient-based optimizations.

In this paper, we focus on deep neural network (DNN) architectures, built from layers representing affine-linear transformations parametrized by the parameters $\boldsymbol{\theta}$. Throughout this work, we restrict the discussion to fully-connected layers, and convolutional layers. Different layers $l$ are interlaced with nonlinear activation functions $f_l(\cdot)$ for enhanced expressivity. The output of the first layer $z$ involves no bias ($\boldsymbol{b}_i^{(1)} \equiv 0$, see below) and is always fed into



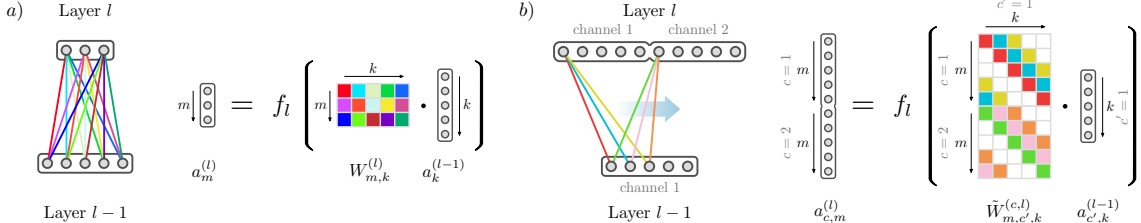

Figure 1: Schematic depiction of the network layers and visualization of the corresponding mathematical meaning. a) Dense layer, cf. Eq. (13). Each neuron in layer $l$ is connected to all neurons in layer $l-1$ without further structure. b) Convolutional layer, cf. Eq. (26). The connectivity is typically sparse and within a channel the coupling of each neuron to the previous layer is obtained by translation as indicated by the arrow and evident from the structure of the coupling matrix. In both cases we omitted possible additional biases that can be added to each layer.

an even activation function $f_1(z) = \ln \cosh(z)$ to incorporate spin inversion symmetry (see App. C). The subsequent layers utilize the odd activation $f_{l>1}(z) = z - \tanh(z)/2$, which is linear around $z = 0$. This was empirically found to be advantageous for learning, because it allows to alleviate the vanishing gradient problem for deep networks [78, 97].

A single fully-connected layer $l$ is defined as,

$$z_i^{(l)} = \sum_j W_{ij}^{(l)} a_j^{(l-1)} + b_i^{(l)}, \qquad a_i^{(l)} = f_l(z_i^{(l)}), \tag{13}$$

with variational parameters $\boldsymbol{\theta}^{(l)} \equiv \{W^{(l)}, b^{(l)}\}$, where $W_{ij}^{(l)}$ is known as the weight matrix and $b_i^{(l)}$ is the bias vector. This layer is pictured in Fig. 1 (a). General fully connected layers such as Eq. (13) can be supplemented with additional structure to render the ansatz better-suited for the problem of interest. For instance, often times in physical systems the final result is expected to have spatially local and translationally invariant structures, which can be more easily captured by a convolutional layer instead, see Fig. 1 and App. B.

In the exponentially large Hilbert spaces of many-body systems, the ratio of (squared) amplitudes required for Monte Carlo sampling, see Eq. (5), can differ by several orders of magnitude. Therefore, it is more convenient to work directly with $\ln \psi_{\boldsymbol{\theta}}(\boldsymbol{s})$ instead of $\psi_{\boldsymbol{\theta}}(\boldsymbol{s})$. We identify the network input with the spin configuration $\boldsymbol{s}$, i.e, $a_j^{(0)} \equiv s_j$, and choose the output layer $N_L$ to contain a single neuron representing the log-amplitude of the variational wavefunction, $a^{(N_L)} \equiv \ln \psi_{\boldsymbol{\theta}}(\boldsymbol{s})$. The network architectures we use in subsequent simulations are described in App. H.

Before we start each simulation, we choose uniformly distributed initial parameters $\boldsymbol{\theta}_k$, see App. D. This results in a uniform log-amplitude distribution and an approximately delta-peaked phase distribution for all spin configurations. Hence, one can convince oneself that the corresponding physical initial state is the (non-normalized) $S^x$-polarized state $|\phi\rangle \propto \bigotimes_{j=1}^{N} \left( |\uparrow\rangle_j + |\downarrow\rangle_j \right)$, up to an overall phase factor. Because the $J_1 - J_2$ model is SU(2) symmetric (see App. C), this state is equivalent to the ferromagnetic $S^z$-polarized state, corresponding to the most excited eigenstate in the spectrum of the Hamiltonian (1). Thus, the energy density of the physical initial state amounts to $\langle \phi | \mathcal{H} | \phi \rangle / N = (J_1 + J_2)/2$, and the energy variance vanishes.

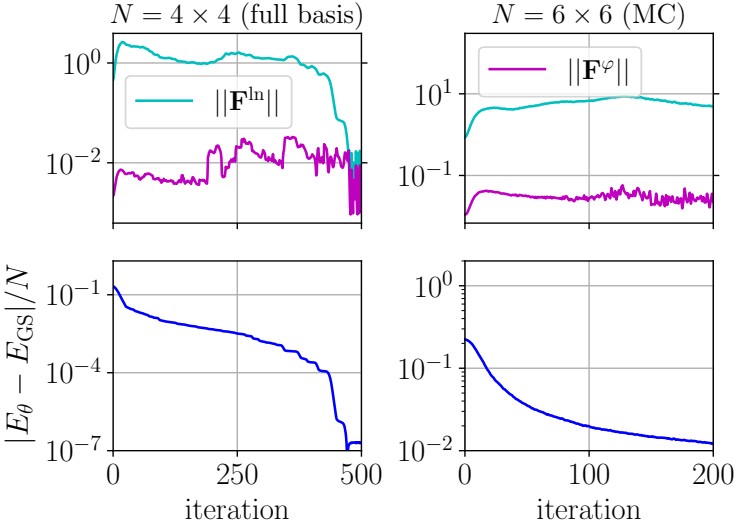

Figure 2: Top row: the norm of the force vector $||\boldsymbol{F}_k||$ separated into a log-amplitude and phase contributions shows the large difference between the two in the initial stages of optimization. Bottom row: the corresponding energy optimization curves. Left column: full-basis simulation for $N = 4 \times 4$. Right column: C simulation at $N = 6 \times 6$ with $N_{\mathrm{MC}} = 2^{15}$ samples. The log-amplitude contribution to the gradients dominates over the phase contribution at the early optimization stages: since the two contributions enter additively in the network update vector, this might lead to an optimization bias, and prevent learning the correct sign structure at $J_2/J_1 = 0.5$ (until very deep in the optimization landscape). The AF Marshall-Peierls rule is incorporated into the ansatz through a gauge choice. The network architecture used is listed in App. H. Optimization was done using RK in combination with SR, see App. A.

## 3 Numerical instabilities

In the following, we elaborate on several practical problems that arise in the implementation of variational neural network wave functions. In particular, the approach suffers from numerical instabilities that require special attention. We further contrast the choice of a single holomorphic network to encode the wave function, with representing the amplitude and phase separately using two independent real-valued networks. We argue that the holomorphic network comes with two major drawbacks: (i) restrictions in learning the non-trivial sign structure of the wave function due to the holomorphic constraint on gradients, and (ii) inherent instabilities caused by the unbounded character of holomorphic activation functions. Whereas (i) can be taken care of by working with two independent real-valued networks, we exhibit the necessity to introduce a regularization layer for the log-amplitude network to eliminate otherwise fatal generalization errors that occur during training, even with bounded activation functions.

Although a real-valued Hamiltonian has real-valued eigenstates, in the computational basis $\{s\}$, the wavefunction coefficients $c_s$ can have both positive and negative signs, see Eq. (2). In this work, we focus on the following two possibilities to encode the logarithmic wave function coefficients, see Sec. 2.2: we either use a holomorphic network with complex parameters (Sec. 3.1), or two independent real networks for phase and amplitude (Sec. 3.2).

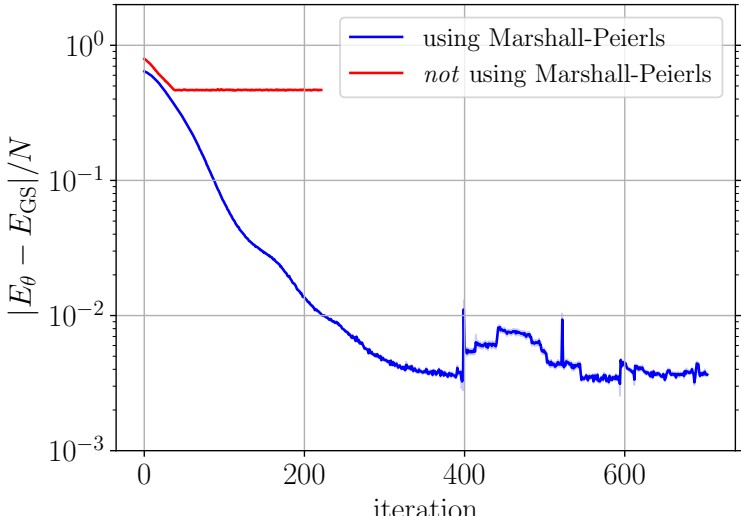

Figure 3: A holomorphic single-layer DNN shows a drastically different performance, depending on whether the AF Marshall-Peierls sign rule is used (blue) or not (red). The spikes in the blue curve after iteration 400 occur due to a run-away instability related to poor generalization, see Sec. 3.1.2. The system size is $N = 6 \times 6$. Optimization was done using SR with the SGD optimizer, see App. A, with fixed learning rate $\gamma = 10^{-2}$ and $N_{\mathrm{MC}} = 2^{15}$ MC samples. The network architecture is listed in App. H, and $J_2/J_1 = 0.5$.

## 3.1 Holomorphic neural quantum states

A distinguished class of complex-valued functions are the holomorphic functions. Holomorphic neural states are defined by complex-valued parameters $\boldsymbol{\theta} \in \mathbb{C}$, coupled by holomorphic activation functions $f_l(\cdot)$, e.g., $f_l(\cdot) = \ln \cosh(\cdot)$. By definition, such an ansatz can only encode holomorphic approximations to the wavefunction amplitudes. Viewed over the field of real numbers, a key feature of holomorphic functions is that they obey the Cauchy-Riemann equations [98]. This constraint reduces by a factor of two the number of independent derivatives computed in backpropagation, as compared to non-holomorphic functions, which results in a "holomorphic correlation" between the real and imaginary parts of the variational parameters.

### 3.1.1 Holomorphicity-induced correlations between amplitude and phase gradients

In Sec. 2.2, we argued that it is advantageous to approximate directly the logarithm of the variational many-body wavefunction. The polar decomposition, $\ln \psi_{\boldsymbol{\theta}}(\boldsymbol{s}) = \ln|\psi_{\boldsymbol{\theta}}(\boldsymbol{s})| + i\varphi_{\boldsymbol{\theta}}(\boldsymbol{s})$, induces a corresponding decomposition in the force vector $\boldsymbol{F}_k = \boldsymbol{F}_k^{\ln} + \boldsymbol{F}_k^{\varphi}$ of Eq. (9) as follows,

$$\boldsymbol{F}_k^{\ln} = \sum_{\{\boldsymbol{s}\}} p_{\boldsymbol{\theta}}(\boldsymbol{s}) \left( \partial_{\theta_k} \ln|\psi_{\boldsymbol{\theta}}(\boldsymbol{s})| \Big[ E_{\boldsymbol{\theta}}^{\mathrm{loc}}(\boldsymbol{s}) - E_{\boldsymbol{\theta}} \Big] \right),$$

$$\boldsymbol{F}_k^{\varphi} = -i \sum_{\{\boldsymbol{s}\}} p_{\boldsymbol{\theta}}(\boldsymbol{s}) \left( \partial_{\theta_k} \varphi_{\boldsymbol{\theta}}(\boldsymbol{s}) \Big[ E_{\boldsymbol{\theta}}^{\mathrm{loc}}(\boldsymbol{s}) - E_{\boldsymbol{\theta}} \Big] \right). \tag{14}$$

This allows us to separately trace back the energy gradient contributions from the amplitude and the phase of the trial wavefunction. In Fig. 2, we display the norm of the force vectors of Eq. (14) as a function of the optimization step for a two-layer holomorphic neural network.

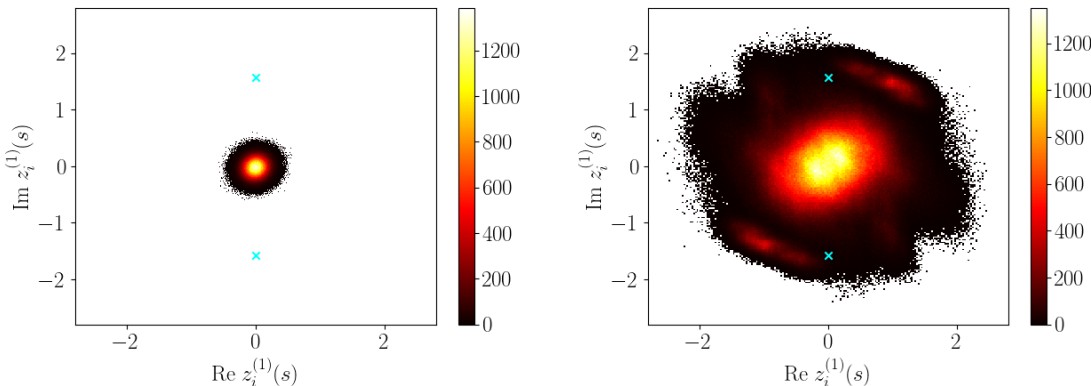

Figure 4: Snapshots of the 2D histogram formed by the complex-valued output $z_i^{(1)}$ of the first layer just before applying the nonlinearity in a holomorphic DNN for two fixed training iterations (iteration 1: left, and iteration 200: right) show the spread of the gradual output across the complex plane. Eventually, a singularity (marked by a cyan cross for the case of $f = \ln \cosh$) is inevitably hit which produces divergent gradients leading to a training instability. To generate the data, we used the holomorphic DNN from Fig. 2 at $N = 6 \times 6$, and sampled 1000 spin configurations at the given training iteration which are then symmetrized and fed back into the same network.

The significant difference of two orders of magnitude observed between the two contributions suggests that the variational optimization is initially dominated by changes in the amplitudes.

For $N = 4 \times 4$, we perform a full-basis simulation which converges easily to the ground state. The magnitude of the log-amplitude contribution eventually becomes of the same order as the magnitude of the phase contribution only deep in the optimization landscape. It is at this later stage that the network starts learning the correct phase distribution $\varphi_s$, which we can verify by comparing to the sign structure of the ground state wave function obtained from exact diagonalization.

For $N = 6 \times 6$, using Monte Carlo sampling, we observe that the phase contribution remains small in the first stages of the optimization. It is an open question whether this behavior persists to the later stages of training, or for holomorphic networks with more parameters. We were not able to perform longer simulations due to the instabilities discussed in the following Sec. 3.1.2. While our data does not predict whether this effect is a generic, Hamiltonian-independent feature, this observation raises a flag to keep in mind when using holomorphic neural states to approximate non-stoquastic quantum states.

It is interesting to note that, when it does not have the antiferromagnetic Marshall-Peierls sign rule built-in, the holomorphic ansatz fails to find a good approximation to the ground state even for deeper networks, see Fig. 3. We believe that this behavior arises due to the holomorphicity-induced constraint on the network parameters. That said, note that *partially holomorphic*, symmetry-restoring RBM's with the antiferromagnetic Marshall-Peierls sign rule built-in, have recently been reported to outperform convolutional neural networks [80].

### 3.1.2 Numerical instabilities in holomorphic neural networks

The domain of the holomorphic activation functions (or non-linearities) is the entire complex plane, as there are no other constraints on the variational parameters $\boldsymbol{\theta}$. In complex analysis, Liouville's boundedness theorem states that all non-constant entire functions are necessarily unbounded [98]. We find that this can potentially lead to two kinds of numerical instabilities.

First, holomorphic activation functions, such as $f(z) \equiv \ln\cosh(z)$ which was originally introduced in the context of restricted Boltzmann machines [58, 83, 99–101], have poles in the complex plane. We have observed that the optimization dynamics can tune the variational parameters in such a way as to hit exactly the singularity, causing the algorithm to "blow up". To demonstrate the mechanism behind this instability, consider the holomorphic DNN trained in Fig 2 [right panel] for $N = 6 \times 6$. At a fixed training iteration, we can sample a batch of spin configurations from the DNN; we then symmetrize them [cf. Sec. C], and feed them back into the same network, only this time we cut the network open and consider the output $z_i^{(1)}$ after the first dense layer, before applying the nonlinearity $f_1$. In order to monitor the spread of the pre-activations $z_i^{(1)}$ across the complex plane, we construct a two-dimensional histogram over the pairs of real and imaginary parts of $z_i^{(1)}$, evaluated at the symmetrized sample. Figure 4 shows snapshots of such histograms at training iterations 1 [left] and 200 [right], which exhibit the behavior that we typically found: The pre-activations $z_i^{(1)}$, which take small absolute values at early iterations, increase in magnitude during training. Nonetheless, the distribution remains dense, meaning that poles – if present – are eventually inevitably encountered. In Fig. 4 we marked exemplarily two poles of the holomorphic nonlinearity $f_1(\cdot) = \ln\cosh(\cdot)$ at $z^* = \pm i\pi/2$ [cyan crosses]. A single pre-activation $z_i^{(l)}$ lying sufficiently close to a pole for just a single input sample is sufficient to cause a divergent gradient contribution and thereby a training instability. For this reason, the complex networks used in this paper [including the ones in Fig 2], are trained using fourth-order polynomial nonlinearities.

Although entering poles in the complex plane can be partially alleviated by using analytic functions instead (e.g., a polynomial approximation to $f(z)$ for small $|z|$ as proposed in Ref. [78]), the unbounded character remains. This naturally leads to a second kind of instability, triggered by a runaway effect: since the neural quantum states we consider are not normalized (and cannot be, for practical reasons in large many-body systems), the magnitude of the log-amplitude $\ln|\psi_\theta(s)|$ can increase indefinitely. As a consequence, one often encounters spin configurations during the optimization procedure with an incorrect estimate of the ratio $|\psi_\theta(s')|/|\psi_\theta(s)|$ by a few orders of magnitude. This leads to a high variance in the Monte Carlo estimate of the local energy [Eq. (4)] or, alternatively, to artificial spikes in the probability distribution $p_\theta(s)$. The resulting incorrect estimates of the force vector $F_k$ of Eq. (9) first cause the algorithm to update the parameters using incorrect gradients, and then eventually to blow up. We find that this issue can often be remedied by using an adaptive learning rate solver, such as Runge-Kutta (RK), which controls the learning rate schedule, see App. A.

Finally, let us emphasize that the representativity theorems for neural networks to be universal approximators in the limit of infinitely many neurons explicitly rely on the boundedness of the activation functions [85, 86]. This provides yet another motivation for us to explore alternatives, like those introduced in the following section.

## 3.2 Complex-valued neural quantum states with decoupled real-valued networks

Our previous considerations motivate us to consider two independent real-valued networks to model the complex-valued wave function, $\psi_\theta(s) = |\psi_{\theta^{(1)}}(s)|e^{i\varphi_{\theta^{(2)}}(s)}$ with $\theta \equiv (\theta^{(1)}, \theta^{(2)})$: one network approximates the log-amplitudes, $\ln|\psi_{\theta^{(1)}}(s)|$, and the other encodes the phases, $\varphi_{\theta^{(2)}}(s)$. A similar ansatz was recently constructed using long-range entangled plaquette states [102]. Importantly, this ansatz allows for the gradients of the network parameters to be evaluated separately: the parameter optimization following Eqs. (10) and (11) can be computed independently for the amplitudes and the phases. Although the log-amplitude and phase networks are independent, the local energy of Eq. (4), and therefore also the values of the network gradients, depend on the output of *both* the log-amplitude and the phase networks.

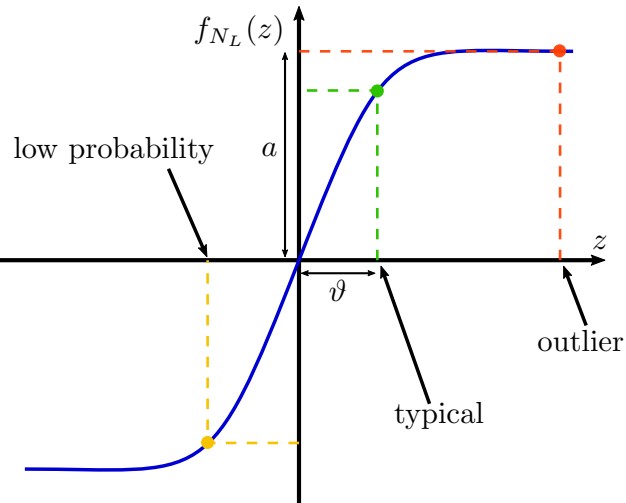

Figure 5: Schematic depiction of the role of the regularizing layer at later stages of optimization. Typical configurations are mapped to the upper end of the linear regime, which allows for maximal contrast to unlikely configurations. As a result outliers with otherwise strongly overestimated amplitudes due to faulty generalization are regularized, because they end up in the flat tail of the activation function.

The optimization of networks with real-valued parameters proceeds similar to Eqs. (10) and (11). The equation for gradient descent remains the same. However, in the Stochastic Reconfiguration update we use directly the real parts of the $F_k$ and $S_{kk'}$,

$$
\begin{aligned}
\boldsymbol{\theta}_k &\longleftarrow \boldsymbol{\theta}_k - 2\gamma\mathrm{Re}(\boldsymbol{F}_k), \\
\boldsymbol{\theta}_k &\longleftarrow \boldsymbol{\theta}_k - \gamma\sum_{k'}\mathrm{Re}(\boldsymbol{S}_{kk'}^{-1})\mathrm{Re}(\boldsymbol{F}_{k'}).
\end{aligned}
\tag{15}
$$

Real-valued networks can be constructed using bounded non-linearities. This automatically resolves the instability caused by the poles of the activation function discussed in Sec. 3.1.2. Besides, the runaway instability can be mitigated by introducing a single bounded layer at the top of the amplitude network. Denoting the symmetrized network output in the last layer as $z$ [cf. App. C], we apply to it the activation function,

$$
f_{N_L}(z) = a\tanh\left[(z-\vartheta)/a\right] + \vartheta,
\tag{16}
$$

for a fixed parameter $a$. Only $\vartheta$ is an additional variational parameter here. Setting $a = 8$ results in a total log-amplitude network output range of 16, and is chosen empirically to allow for the log-amplitude network to encode a maximum relative magnitude difference of $|\psi_{\boldsymbol{\theta}}(\boldsymbol{s}')|/|\psi_{\boldsymbol{\theta}}(\boldsymbol{s})| \approx 10^7$. The activation function $f(z)$ in Eq. (16) was chosen to have a linear slope for small values of $z$. Because the input for this layer is $|z| \ll 1$ in the first few optimization iterations, $f(z)$ has no effect in this early stage of the optimization dynamics. Later on, the parameter $\vartheta$ will automatically adjust the position of the linear regime in $f(z)$ relative to typical values of $z$, which allows to cut off the large values $\ln|\psi_{\boldsymbol{\theta}}(\boldsymbol{s})|$ for those configurations $\boldsymbol{s}$ causing the instability.

During the optimization we observed that the parameter $\vartheta$ is adjusted such that the preactivation of typical configurations lies close to the upper end of the linear regime, allowing to represent accurately their relation to configurations less likely to occur in the MC sample, as depicted in Fig. 5. Thereby, outliers with a too large pre-activation are automatically regularized as they end up in the flat tail of the $\tanh(\cdot)$ function. We checked that promoting $a$ to

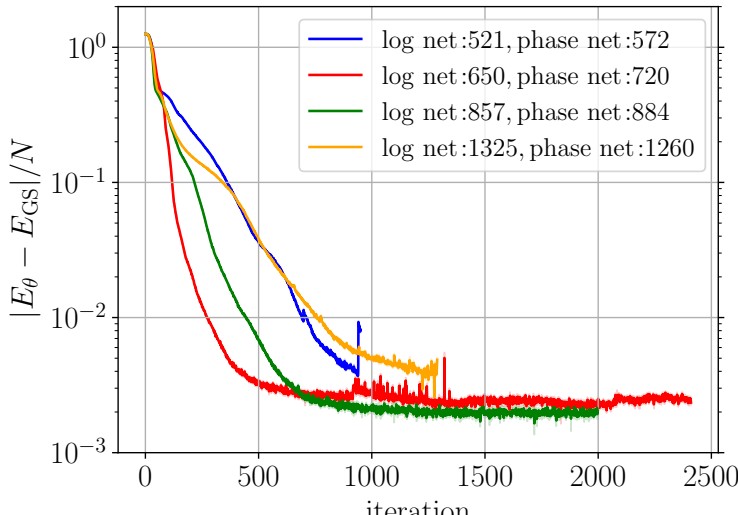

Figure 6: Variational energy density against the iteration number for a fixed neural network layer structure with increasing number of parameters (neurons). The system size is $N = 6{\times}6$. The network architectures used are listed in App. H, and $J_2/J_1 = 0.5$. Optimization was done using RK in combination with SR, see App. A. A total of $N_{\mathrm{MC}} = 2^{15}$ MC samples were used.

be a variational parameter itself produces worse results since this allows for a variable output range of the log-amplitude network and the runaway instability eventually kicks in.

We emphasize that using a bounded activation function alone is not sufficient: if inaccurate updates to the network parameters are generated, e.g., due to a large learning rate, this instability may occur in milder forms, as visible in Fig. 6 (blue line) and also in Fig. 15 in App. A for the SGD/SR optimizer. Nonetheless, such a scenario can further be prevented by using an adaptive learning rate algorithm, see App. A, and a sufficiently large MC sample size required for accurate gradient estimates.

The runaway instability only affects the log-amplitude network since the values of the phase network end up winding in the argument of the exponential $e^{i\varphi_{\theta^{(2)}}(s)}$: this reflects the known fact that only phase differences in the interval $[0, 2\pi)$ are physical. In fact, we observed that the optimization procedure makes use of this gauge freedom towards the later stages of optimization, to position the values $\varphi_{\theta^{(2)}}(s)$, corresponding to certain configurations $s$, in different Riemann sheets at a distance $2\pi\ell$ apart, with $\ell \in \mathbb{Z}$. Hence, we find no merit in using a discontinuous $\arg(z)$ activation function [64] in the output layer of the phase network, which may cause further problems when computing gradients using backpropagation. Since physical observables are only sensitive to phase differences, this phase network ansatz can result in constant relative phase shifts from one iteration to the next, observed at the later stages of the optimization.

For a fixed system size $N = 6 \times 6$, we display in Fig. 6 the variational energy density as a function of the optimization iteration step for different numbers of variational parameters and a fixed architecture consisting of three convolutional layers followed by two fully-connected layers. We find that the optimization algorithm does not consistently yield a better variational approximation when increasing the network size[1]. Hence, our data indicates that the current

---

[1]For the selected hyperparameters, the blue curve failed the samples post-selection checks, see App. F, after ten consecutive attempts of drawing a new MC sample, but the other simulations were well-behaved.

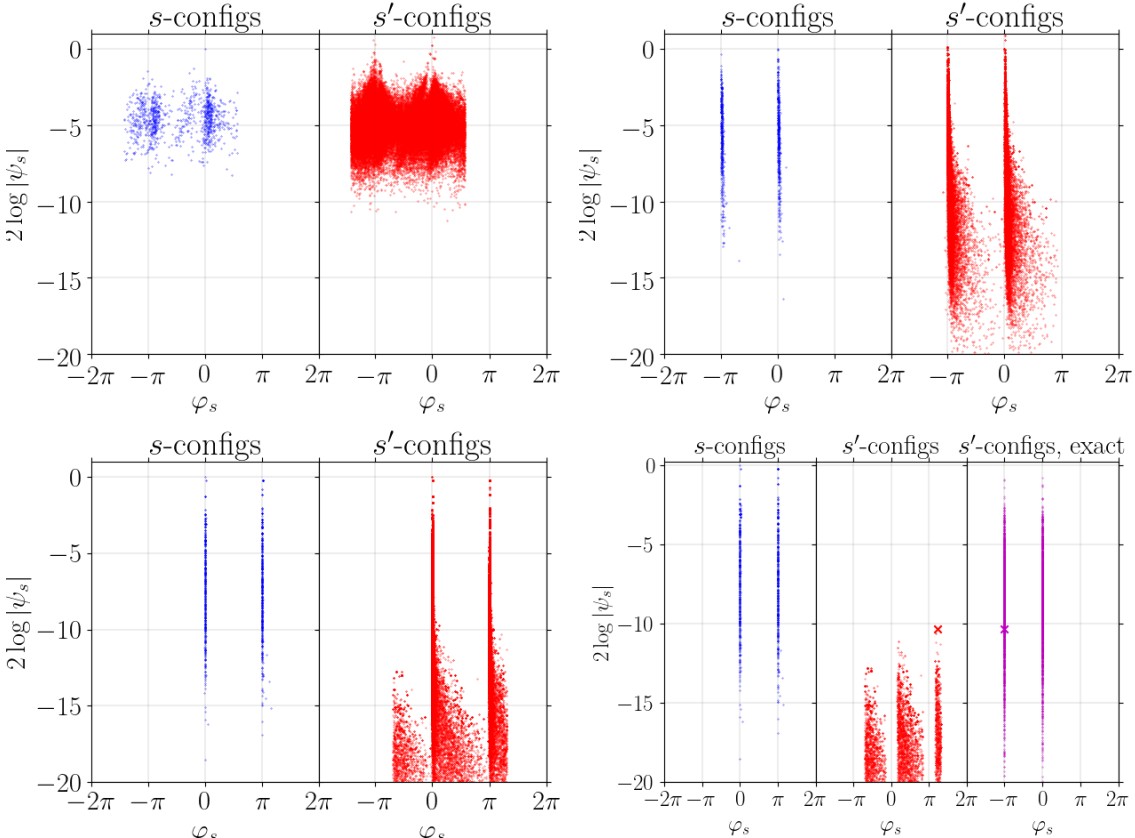

Figure 7: Snapshots of the distribution of phases and magnitudes at increasing optimization iterations during training (top left: 200, top right: 500, bottom left: 1995) shows the emergent of two phase peaks, a distance $\pi$-apart, signaling the approach to the real-valued ground state wave function (see text). Left panels (blue): a $\boldsymbol{s}$-sample of $10^3$ configurations drawn from the log-amplitude network. Right panels (red): spin configurations that contribute to the local energy and have a nonzero Hamiltonian matrix element ($\boldsymbol{s}'$-sample). Bottom right: at iteration 1995 subtracting the peaks from the $\boldsymbol{s}'$-sample (middle) in an $\varepsilon = 0.5$ vicinity of $\varphi_s = 0, \pm\pi$ leaves only the mismatch configurations (red, middle), whose amplitudes and signs are compared against the exact ground state values (right, magenta). The position of the largest-amplitude $\boldsymbol{s}'$-configuration is held fixed and is denoted by a red cross. The system size is $N = 6 \times 6$. The network architecture is listed in App. H, and $J_2/J_1 = 0.5$. Optimization was done using RK in combination with SR, see App. A. The MC sample size is $N_{\mathrm{MC}} = 2^{15}$. Note that the values on the $y$-axes are not absolute, since neural quantum states are not normalized.

approach does not produce a well-controlled variational approximation to the ground state wave-function at $J_2/J_1 = 0.5$. However, as we argue in Secs. 5 and 6, this observation is not related to the ability of neural quantum states to approximate the ground state (the so-called ansatz expressivity), but is rather an intrinsic property of the rugged variational landscape at $J_2/J_1 \approx 0.5$.

# 4 Learning the phase structure

For real-valued ground state wavefunctions, the correct sign distribution, i.e., $\text{sign}[\psi_{\boldsymbol{\theta}}(\boldsymbol{s})]$, is difficult to find for frustrated systems in the $S^z$ basis. In Sec. 3.1, we introduced complex-valued variational wavefunctions for which learning the correct signs correspond to learning the correct phase distribution $\varphi_{\boldsymbol{\theta}^{(2)}}(\boldsymbol{s})$.

In this section, we reveal some of the difficulties associated with learning the sign structure of the ground state wavefunction in the $J_1 - J_2$ model. First, we analyze data obtained from the neural network states, and show that it does not encode the exact ground state sign distribution; we then briefly quantify how much the exact ground state sign distribution differs from the Marshall-Peierls distribution.

In Fig. 7, we show snapshots of the phase distribution during the training process as a scatter plot in the $\ln|\psi_{\boldsymbol{\theta}^{(1)}}|^2$ versus $\varphi_{\boldsymbol{\theta}^{(2)}}$ plane. The left panels (blue data points) show a sample of $10^3$ spin configurations $\boldsymbol{s}$ drawn according to the probabilities encoded in the log-amplitude network at a fixed training iteration [see caption]. The right panels (red data points) show all configurations $\boldsymbol{s}'$ which contribute a nonzero matrix element $\mathcal{H}_{\boldsymbol{s}\boldsymbol{s}'}$ to the local energy, see Eq. (4) (these are about $8 \times 10^4$ configurations for $N = 6 \times 6$ at the largest training iteration shown). Since phases are defined modulo $2\pi$, we wrap the output of the phase net in the interval $[-\pi, \pi)$. To fix the global phase, we find the configuration $\boldsymbol{s}_0$ in the $\boldsymbol{s}$-sample of $10^3$ states with the largest value of $\ln|\psi_{\boldsymbol{\theta}^{(1)}}(\boldsymbol{s})|$, and use it to set $\ln|\psi_{\boldsymbol{\theta}^{(1)}}(\boldsymbol{s}_0)| = 0$ and $\varphi_{\boldsymbol{\theta}^{(2)}}(\boldsymbol{s}_0) = 0$.[2]

In Fig. 7, we observe two emerging peaks along the $\varphi$-axis at a distance $\pi$ apart. Hence, we see that the algorithm correctly identifies that the ground state is real-valued (up to a global phase). While the phases of all states of the $\boldsymbol{s}$-sample are learned to be separated by $\pi$ at the later training iterations, there exist small-amplitude $\boldsymbol{s}'$-states which are misaligned with the main phase peaks. This can be explained by noting that these states have tiny amplitudes and do not contribute significantly to the local energy, and hence also to the gradients used to update the variational parameters[3].

To investigate whether our network learned the correct sign distribution, we do a comparison against the exact ground state, obtained using exact diagonalization on a $6 \times 6$ lattice [1, 103, 104]. At a fixed iteration, for each set of configurations (the blue and red data points in Fig. 7), we evaluate the signs according to (i) the antiferromagnetic Marshall-Peierls sign rule, (ii) the exact ground state at $J_2/J_1 = 0.5$, and (iii) the optimized neural network at the latest available iteration (Fig. 7, lower right panel):

- Out of the $10^3$ samples $\boldsymbol{s}$, 99.8% have identical signs in the AF Marshall-Peierls rule and the exact ground state. This means that the large majority of states which can be drawn from the network cannot help distinguish the true ground state phase structure from the AF Marshall-Peierls signs. For the remaining 0.2% mismatch configurations, we computed the neural network predictions and compared it against AF Marshall-Peierls and the exact ground state distribution. We find a clear agreement with the AF Marshall-Peierls rule, implying that the neural network did not learn the correct sign distribution,

---

[2]Fixing the value of $\ln|\psi_{\boldsymbol{\theta}}(\boldsymbol{s}_0)|$ is allowed because neural quantum states are non-normalized wavefunctions.

[3]A clearly visible feature displayed in the lower row of Fig. 7 is the asymmetry/skewedness of the phase distribution for the samples $\boldsymbol{s}'$. We checked that it represents the manifestation of a dynamical drag effect. It occurs because phases are defined modulo $2\pi$ and the global phase rotates quickly as a result of updates during the training process (one can think of the two phase peaks moving together). Since the large-amplitude states give rise to the major contribution to the updates of the neural network parameters, the low-amplitude states start lagging behind. This drag effect is particularly prominent for those training iterations when the energy changes quickly, or when we do not use an adaptive learning rate solver (e.g., SGD). We checked that the drag can be reduced or completely eliminated by breaking the U(1) gauge invariance associated with the phase network output, e.g., by adding a bounded non-linearity as a final output layer in the phase network. However, this did not improve the variational energies.

even though it was completely unbiased by any pre-training [64] or extra unitary rotations [61].

- In the samples $s'$, the percentage of configurations whose signs do not agree between the AF Marshall-Peierls rule and the exact ground state sign distribution is 14.2%. However, evaluating the neural network on these states showed that the majority of them (82% out of the 14.2%) appeared consistent with the AF Marshall-Peierls rule. Clearly, the accuracy of this assignment decreases for the states with low amplitudes, since they do not belong to a well-defined phase peak.

From these data, we conclude, that our network identifies very accurately the AF Marshall-Peierls rule on high amplitude configurations, whereas at very low amplitudes the coefficients can still have arbitrary phases.

The observations above provide a manifestation of the difficulty of encoding the correct phases in the neural network. It is thus natural to investigate the sign structure of the probability amplitudes in the exact ground state: in the entire computational basis at $N = 6 \times 6$, about 50.22% of all configurations have different signs with respect to the AF Marshall-Peierls rule and the exact ground state. However, all the 50.22% taken together constitute a total of only 1.95% of the norm of the exact ground state. This means that most of these configurations have insignificantly small amplitudes, which makes encountering such states in MC sampling unlikely.

In Fig. 7 [bottom right panel], we compare the variational amplitudes of the neural network configurations which do not align well with the main phase peaks, to their exact ground state values. To do this, we consider again the $s'$-sample (red data) at training iteration 1995, and define an $\varepsilon$-vicinity around the peaks at $\varphi_s = -\pi, 0, \pi$. We now manually remove all $s'$ that fall within the peaks' vicinity. The remaining $s'$-sample clearly shows large deviation from the ground state phase distribution which is known to be real-valued. Evaluating the relative amplitudes of the remaining $s'$-sample in the exact ground state is shown by the magenta data. Note that we cannot compare absolute amplitude values here, since the neural network state is not normalized. However, we can fix the amplitude value of the largest $s'$-configuration as a reference point, see the red cross. The comparison clearly shows that the majority of the misaligned $s'$-configurations have their amplitudes suppressed in the neural network state by a few orders of magnitude, as compared to the exact ground state. Thus, the VQMC optimization procedure conveniently suppresses the weights of those configurations whose phases cannot be learned well and, thus, their contribution to the gradient update will also be suppressed. As a result, such configurations become difficult to encounter in the MC sample. Since the neural network state in VQMC improves from data generated by the network itself, this slows down the energy minimization procedure.

To sum up, we showed that, instead of learning the exact ground state sign distribution, starting with no bias our neural network approximates closely (though not exactly) the Marshall-Peierls sign rule. We traced the reason for this back to the vanishingly small weights that non-Marshall-Peierls configurations have in the exact ground state, which suppresses their occurrence in the MC samples used for training.

# 5 Identifying bottlenecks which prevent learning the ground state

The results we report raise the obvious question as to *what* prevents learning the correct ground state. Pinpointing the major issue(s) is important from the perspective of improving the neural quantum state VMC technique. For $N = 6 \times 6$, we can compute the exact ground state wavefunction using exact diagonalization, and use it to design a few numerical experiments.

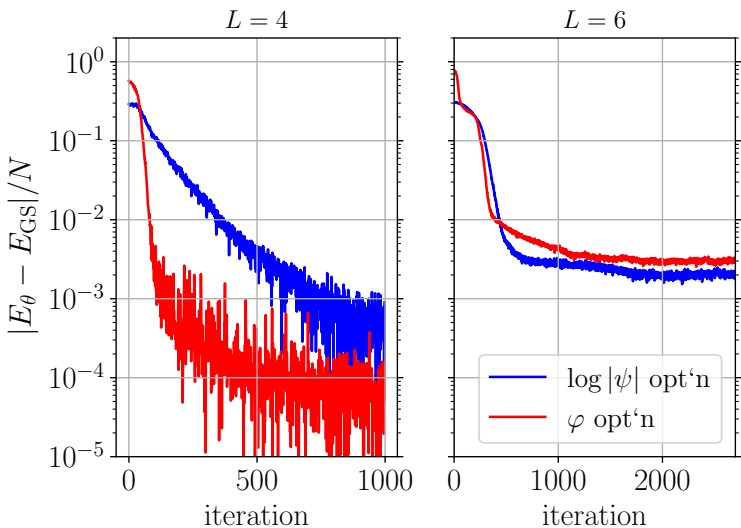

Figure 8: Energy minimization in the partial learning problem for $N = 4 \times 4$ (left panel) and $N = 6 \times 6$ (right panel): $\ln|\psi|$-optimization (blue) shows the amplitude learning curves for the log-amplitude network, using the sign structure of the exact ground-state at every iteration step (i.e., we do not use a phase network). $\varphi$-optimization (red) shows the phase learning curves for the phase network, using the amplitudes of the exact ground state at every iterations step (i.e., we do not use a log-amplitude network). The network architecture is listed in App. H, and $J_2/J_1 = 0.5$. Optimization was done using RK in combination with SR, see App. A. The MC sample sizes are $N_{\mathrm{MC}} = 2^{10}$ for $N = 4 \times 4$, and $N_{\mathrm{MC}} = 2^{15}$ for $N = 6 \times 6$.

Our goal is to check if the training bottleneck arises due to

- Problems with the optimization of either the log-amplitude or the phase network alone,

- A MC sampling issue related to the probability of encountering states that can produce the correct gradients to minimize the energy,

- The expressivity of the log-amplitude or phase network,

- An issue with the optimization procedure/algorithm.

To this end, we perform two sets of experiments, namely studying partial learning problems, where only either phase or amplitude has to be learned (Sec. 5.1), and eliminating Monte Carlo noise by performing full basis simulations (Sec. 5.2). For a fair comparison, we use the same neural network architecture employed in the majority of the figures throughout the paper.

## 5.1 The partial learning problems

Consider first the two partial learning problems: (i) sampling from the exact ground state distribution $|\psi_{\theta}(s)|^2$, we optimize only the phase network, and (ii) given the correct values for the phase distribution in the exact ground state, we optimize only the log-amplitude network to see if we can learn the magnitudes of the ground-state probability amplitudes.

Figure 8 shows the energy optimization curves for $N = 4 \times 4$ (left panel) and $N = 6 \times 6$ (right panel). Scenario (i) is shown in red and scenario (ii) in blue. As anticipated, we see

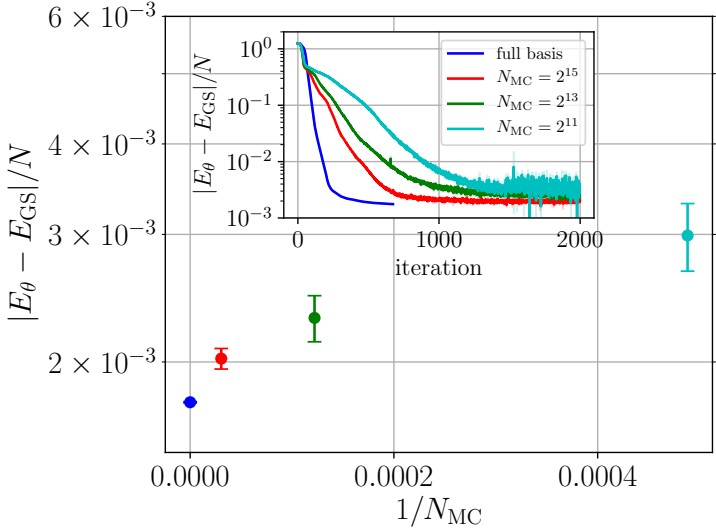

Figure 9: Lowest variational energy density reached, and its MC error bar versus the inverse size of the MC sample $1/N_{\mathrm{MC}}$: increasing the sample size does not necessarily lead to a lower energy. Inset: variational energy density against the iteration number for different sample sizes; see inset legend for the values of $N_{\mathrm{MC}}$. The system size is $N = 6 \times 6$. The network architecture is listed in App. H, and $J_2/J_1 = 0.5$. Optimization was done using RK in combination with SR, see App. A.

that the optimization procedure results in energy difference with the exact ground state below $10^{-3}$ for $N = 4 \times 4$ (left panel). However, for $N = 6 \times 6$ we find similar behavior as in the full learning problem: (i) optimizing the phase network by sampling from the exact ground state probability distribution, the system ends up trapped in a plateau, corresponding (largely) to the AF Marshall-Peierls rule, which survives at least up to 4000 optimization cycles (not shown). Importantly, the value of the plateau appears higher than that obtained in the full-learning problem for the two independent simulations we ran. (ii) Interestingly, although optimizing the log-amplitude network given the correct phases, leads to a slightly lower energy, as compared to (i), it does not perform much better than the joint learning problem.

This numerical experiment implies that the log-amplitude/phase-network has trouble learning the correct ground state on its own, even if the second network is taken to produce the exact ground state data. Hence, it remains unclear if it is the phases alone that prevent the algorithm from reaching the ground state (for otherwise, scenario (ii) would reach the ground state).

## 5.2 Full-basis simulation

To test whether the failure to learn the correct phases is caused by the MC sampling, we can turn it off, and work with the full basis of 15 804 956 states[4] in the ground state symmetry sector at $N = 6 \times 6$ (excluding SU(2) symmetry, see App. C). We use the same neural network architecture as before, but now all states in the Hilbert space contribute to the evaluation of the gradients $\boldsymbol{F}$ and the curvature matrix $\boldsymbol{S}$, i.e., we perform an exact simulation, free of MC sampling noise, which we refer to as a *full-basis simulation*[5]. In Fig. 9, we fix the seed of the

---

[4]For comparison, at $N = 4 \times 4$, the Hilbert space dimension of the ground state sector is 107.

[5]Performing a full-basis simulation is feasible using massive parallelization, and automated differentiation software [105], and to the best of our knowledge, it has not been performed so far on this problem for large Hilbert

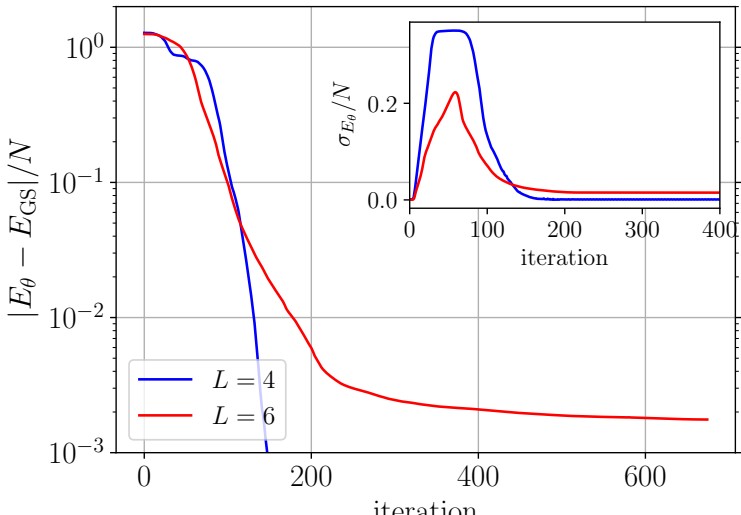

Figure 10: Energy minimization in the full-basis simulation learning problem for $N = 4 \times 4$ (blue) and $N = 6 \times 6$ (red). Albeit faster, for $N = 6 \times 6$, we obtain similar energies as with MC sampling which implies that the problem with reaching the correct ground state is not (directly) related to MC sampling. The network architecture is listed in App. H, and $J_2/J_1 = 0.5$. Optimization was done using RK in combination with SR, see App. A.

pseudo-random number generator, and compare the energy curves as we vary the size $N_{\mathrm{MC}}$ of the MC sample over two orders of magnitude (including a point free of MC noise, which corresponds to the full-basis simulation). This indicates that MC noise is not the limiting factor in our variational ground state search. Therefore, throughout the rest of this section, we focus on the full-basis simulation.

The full-basis training curves displayed in Fig. 10 show that there is a significant difference in the accuracy achieved, for the two system sizes. While the difference to the ground state energy rapidly drops for the small system[6], the minimal energy obtained for $N = 6 \times 6$ is the same as the one achieved in the previously discussed simulations with Monte Carlo sampling. Therefore, our simulations indicate that the Monte Carlo sampling noise is not the limiting factor that inhibits the optimization procedure to go further down the energy landscape.

Thus, we turn our attention to check the expressivity of the neural network ansatz. This is a particularly challenging task, even if the exact ground state is known, because it requires a definition of expressivity which does not depend on the cost function or the optimization algorithm, for different cost functions or optimizers may yield different results depending on the topography of the optimization landscape. Therefore, we ask the slightly different, yet practically more relevant, question as to whether our best-performing network has reached a local minimum of the energy landscape or not. Ending up in a minimum would be consistent with insufficient expressivity of the ansatz.

To investigate this, we compute the Hessian matrix associated with the energy landscape

$$H_{mn} = \partial_{\boldsymbol{\theta}_m} \partial_{\boldsymbol{\theta}_n} E_{\boldsymbol{\theta}} \,. \tag{17}$$

Negative and positive eigenvalues of the Hessian correspond to directions of negative and positive curvature, respectively. A local minimum has the property that the Hessian is positive

---

spaces.

[6]Minimal energy differences achieved with $N = 4 \times 4$ are of the order of $10^{-7}$, see Fig. 3.

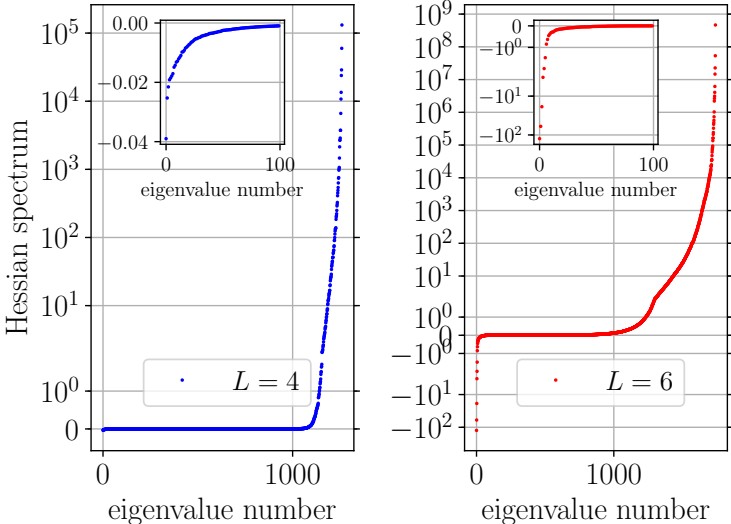

Figure 11: Hessian matrix eigenvalues of the energy cost function, see App. G: full-basis simulation for $N = 4 \times 4$ at iteration 499 where $E_{gs} = -8.457917$ (blue), and $N = 6 \times 6$ at iteration 675 where $E_{gs} = -18.073818$ (red). Inset: zoom over the first 100 eigenvalues. The few negative large eigenvalues at $N = 6 \times 6$ indicates the presence of highly curved narrow directions in the energy landscape, which appear hard for VQMC to find. They are responsible for the slowing down of the rate at which energy improves. The network architecture is listed in App. H, and $J_2/J_1 = 0.5$. Optimization was done using RK in combination with SR, see App. A.

definite, i.e., all its eigenvalues are positive. Albeit a very time-consuming computation, it is feasible to obtain these eigenvalues due to the relatively small sizes of our networks. The details for the derivation of the explicit form of the Hessian matrix for the neural quantum state ansatz is shown in App. G.

Figure 11 shows the Hessian eigenvalues for the $N = 4 \times 4$ and $N = 6 \times 6$ systems at later stages of training. Whereas the $N = 4 \times 4$ Hessian has some negative eigenvalues, note that their magnitude is on the order of $10^{-2}$ (see inset). Therefore, these correspond to flat directions on the variational manifold, that the optimization dynamics easily follows as it continues to improve the energy of the variational state. In stark contrast, the $N = 6 \times 6$ Hessian has a few large negative eigenvalues on the order of $10^2$: these are highly curved sparse directions on the manifold, which are hard to find by the optimizer. This might be causing the apparent slow improvement of the energy curve in Fig. 10. We find that all large negative-curvature directions originate from the log-amplitude network, by computing the contribution of the corresponding eigenvectors to the log-amplitude network parameter subspace (the log-amplitude and phase-network parameters are coupled in the Hessian, see App. G).

More interestingly, the existence of these negative eigenvalues in the $N = 6 \times 6$ system proves that the network parameters have not yet reached a local minimum in the optimization landscape. Hence, expressivity is not a problem causing the observed learning bottleneck. Note that, while this does not imply that our ansatz is expressive enough to encode the exact ground state distribution, it already points at a severe problem with the variational manifold and the optimization dynamics used to explore it.

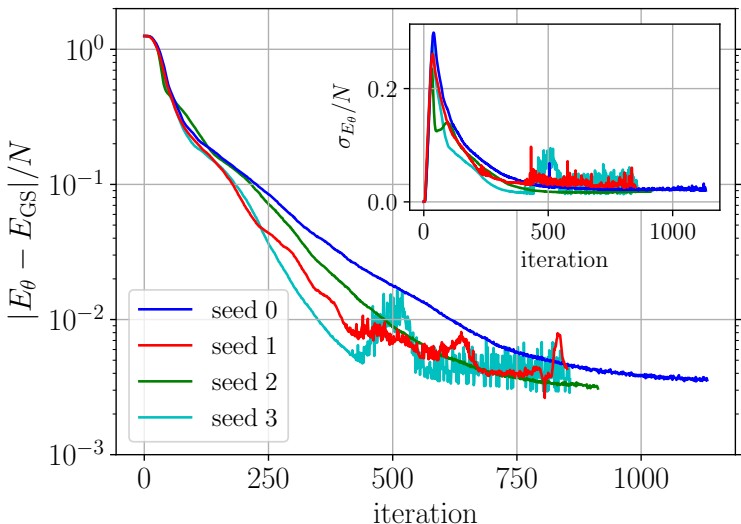

Figure 12: Variational energy density against the iteration number for different seeds of the pseudo-random number generator. Inset: energy variance per spin. The system size is $N = 6 \times 6$. The network architecture is listed in App. H, and $J_2/J_1 = 0.5$. Optimization was done using RK in combination with SR, see App. A. The MC sample size is $N_{\mathrm{MC}} = 10^5$.

## 6 Rugged variational manifold landscape

The ground state search can also be viewed and analyzed as an optimization problem. In this section, we investigate a connection between the properties of the neural network parameter manifold, and the difficulty of learning the correct ground state. We first discuss signatures in the numerical data of the existence of a highly rugged optimization landscape, and then draw several conclusions of both practical and conceptual importance. We restrict the discussion to the $J_1 - J_2$ model, although much of the analysis can be repeated for similar problems.

The difficulties we encountered using variational quantum Monte Carlo to find the true ground state at $J_2/J_1 = 0.5$ already at $N = 6 \times 6$ motivate us to perform another sequence of numerical experiments to unveil some characteristic features of this problem, which imply that we are dealing with a very rugged energy landscape. In particular, we will address the role of pseudo-random numbers, Monte Carlo noise, and finite machine precision.

### 6.1 Different pseudo-random number sequences

Notice first that "randomness" is an important ingredient of VQMC as it occurs in the selection of the MC proposal updates, but also in the choice of the initial values for the neural network parameters $\boldsymbol{\theta}$. In Fig. 12, we fix all hyperparameters of the algorithm, and study its behavior for four different initial seeds of the pseudo random number generator (the inset shows the energy variance, which is zero in the beginning of optimization, since the $S^x$-polarized state is an eigenstate of the Hamiltonian $\mathcal{H}$). Note that the energy optimization curves start deviating already in the first few iteration steps: therefore, we see that different runs of the algorithm follow different trajectories on the parameter manifold, despite the fact that they all start from (almost) the same initial quantum state. Upon closer examination, we notice that some of these trajectories appear to be stable, while others are prone to (runaway) instabilities (Sec. 3.1.2), caused by the optimization steering the variational parameters in a region of

parameter space where the neural network generalizes poorly. Following different trajectories on the parameter manifold is physically irrelevant, provided that the final network parameters represent the same *physical* state. However, we see that, although all simulations gradually minimize the energy, they eventually get stuck in landscape saddles corresponding to *different* physical states, as becomes evident from the different values of the energy reached. [7]

One should keep in mind that the energy landscape in VQMC is not perfectly sharp, i.e., it is only known within the margin of uncertainty produced by the Monte Carlo estimates. MC-induced uncertainty can also cause the optimization to change the trajectory on the parameter manifold. For the $J_1 - J_2$ model, we find that an increasingly better resolution is needed in order to reach lower energies. Moreover, the larger the value of $N_{MC}$, the faster the energy decreases in the initial stages of optimization, see Fig. 9. Once more we observe that different simulations end up in different saddles even when all hyperparameters are held fixed.

Recall that adding small noise (e.g., due to estimating a quantity from sample averages) is a common practice when training machine learning models: for instance, noise helps overcome shallow barriers in the loss function and (in part) motivated the development of *stochastic* gradient descent (SGD). Therefore, it is likely that the different saddles VQMC gets stuck in, are located in deep valleys, separated by high and difficult to overcome energy barriers. This topography is reminiscent of spin glasses, and will be the subject of future studies.

## 6.2 Consequences of finite machine precision

The rugged character of the optimization landscape is best demonstrated in a simple numerical experiment on a $N = 4 \times 4$ lattice where Monte Carlo fluctuations can be eliminated because of the small size of the Hilbert space, by using a full-basis simulation. We consider a shallow holomorphic DNN, and deliberately restrict the number of neurons to four, so finding the exact ground state becomes infeasible (we believe that this regime contains some of the difficulties encountered for larger systems). We then consider two equivalent but different implementations of the same algorithm, both using SGD with SR [cf. App. A]. To remove the remaining uncertainty related to the initial network parameters, we also load the exact same initial parameters for both implementations.

In Fig. 13, we show the result of this toy-simulation: stunningly, we find that the errors accumulated due to finite precision of the machine arithmetic are enough to conceivably alter the trajectory of the optimization dynamics. Initially, the difference between two curves shown in the inset of Fig. 13 remains close to machine precision up to the plateau at 100 iterations, reaching as small as $10^{-15}$, which certifies the correct implementation of the two independent codes. Nonetheless, the difference quickly grows again once the two simulations escape the plateau. Such a plateau is reminiscent of the landscape saddle point featuring a few highly curved directions we discussed in Sec. 5.2. This behavior occurs as the two simulations go down the energy landscape in different directions. These results are disturbing, because they reveal an issue with reproducibility of the simulation. Apparently, it is not enough to keep the same algorithm hyperparameters and the seed of the random number generator fixed: the compute architecture and the low-level software which contains the arithmetic instructions need to also be identical in order to reproduce the data.

The rugged structure of the cost function manifold also explains why different optimizers

---

[7]Notice that deep neural networks are nonlinear function approximators, and thus Eqs. (10) and (11) define nonlinear differential equations. Hence, one may be tempted to relate the observed behavior to classical chaos (trajectories, starting close in phase space eventually end up exponentially far apart). However, this is not the case for imaginary-time evolution, because classical chaos requires that the underlying equations of motion carry a symplectic (i.e., a phase-space) structure. Chaotic manifold dynamics remain an interesting possibility to consider in real-time evolution though.

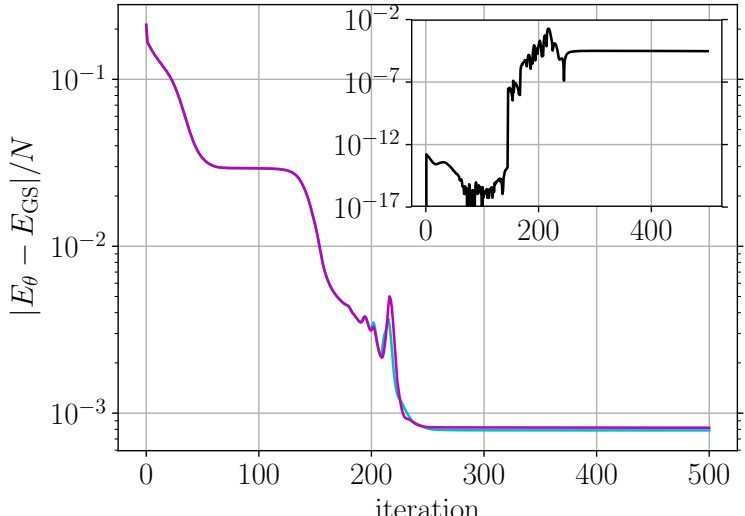

Figure 13: Energy minimization curves in a full-basis simulation. The two simulations (cyan, magenta) use two implementations of the VQMC algorithm with the exact same neural network initial conditions and hyperparameters. Inset (black curve): the difference between the two energy curves grows with increasing the iteration number. We observe a deviation between the two curves deep down the landscape due to an error multiplication seeded by machine precision and caused by the rugged landscape. We used a shallow holomorphic DNN with six hidden neurons (App. H), and $J_2/J_1 = 0.5$. The system size is $N = 4 \times 4$. Optimization was done using SGD in combination with SR, see App. A, with a learning rate of $\gamma = 10^{-2}$.

perform drastically differently, see App. A[8]. This adds to the evidence that the optimization landscape for the ground state search at $J_2/J_1 = 0.5$ features a highly complex topography, consisting of many highly-curved saddles located in deep valleys. It is plausible that the energy landscape is glassy because the problem appears difficult irrespective of the optimization method used (VQMC, tensor networks, etc.). The physical intuition for having a glassy landscape here comes from the frustrated character of the physical Hamiltonian which imposes a number of constraints for the ground state to satisfy in order to minimize energy, akin to $k$-SAT problems in statistical mechanics [106]. Similar results have recently been reported in the context of quantum control [107].

## 7 Conclusions

In this last section, we summarize and discuss our results. In doing so, we pay specific attention to separate conclusions backed-up by numerical evidence from plausible explanations and interpretations of the data. We also discuss some open problems we were not able to resolve. Finally, we give an outlook to relate our study to expected near-term progress in the field.

---

[8]In Fig. 15, it is clear that the optimization dynamics gets stuck in distinct saddles, whose energy values differ by orders of magnitude.

## 7.1 Main results

While they have been demonstrated to work well in a number of problems so far [58,67,72,99], when it comes to learning the ground state of frustrated spin systems, such as the $J_1-J_2$ model, holomorphic architectures for neural network quantum states exhibit certain deficiencies: first, the holomorphic constraint correlates the phase and amplitude gradients of the variational wavefunction parameters, which means that an update to the network parameters will cause a change in both the amplitude and the phase of the output. Therefore, fine-tuning of, e.g., only the phases is not possible using a holomorphic ansatz. Second, non-constant holomorphic non-linearities are necessarily unbounded. This compromises the conditions for the expressivity theorem for neural networks, and raises the question whether holomorphic neural networks can be universal approximators. As a way out, we introduced a natural generalization, which eliminates the above drawbacks, where a complex-valued variational wavefunction is defined by two independent real-valued neural networks: one for the log-amplitudes, and the other for the phases.

Using stochastic reconfiguration to optimize the parameters of neural quantum states can be a challenging problem, due to the occurrence of instabilities, which cause the algorithm to blow up. We identified a runaway instability triggered by unbounded approximations to the probability amplitude: this leads to order-of-magnitude wrong network predictions for the local energies, and, consequently, to wrong values for the gradient updates. The runway instability is related to a poor generalization ability of the network. To remedy it, we proposed to apply a bounded output layer of fixed range, capable of capturing amplitudes which otherwise differ by up to several orders of magnitude. A further instability we discuss in App. D is related to the proper initialization of the neural network parameters, and is particularly relevant for designing deep architectures which do not suffer from the vanishing/exploding gradient problem. Finally, we mention that we found stochastic reconfiguration to produce a much more stable behavior, when combined with an adaptive step-size scheduler, for instance, based on Runge-Kutta methods, see App. A.

Having decoupled amplitude and phase network allowed us to investigate the process of learning the correct ground state phases in an unbiased end-to-end approach, i.e., without any extra physics input. We demonstrated that this removes the necessity to explicitly implement the Marshall-Peierls sign rule in holomorphic networks. Moreover, our phase networks are capable of encoding the Marshall-Peierls sign rule without any pre-training [64], and without the use of a physics-inspired architecture [60, 62, 65]. All simulations (that do not "blow up") end with roughly the same energy density of $E/N \approx -0.5019$. Since the same number occurs in previous studies that differ in the network architecture used and further details of the optimization scheme [61, 62, 64], we believe this behavior to be universal. None of our simulations, including partial-learning and full-basis ones, was able to find the correct phase distribution, or reach a lower energy on the $N = 6 \times 6$ lattice.

On the variational manifold, the Marshall-Peierls sign rule corresponds to a saddle point, where the energy Hessian contains predominantly positive and a few negative eigenvalues. The existence of the negative eigenvalues indicates that the network parameters have not yet reached a local minimum; this shows that the expressivity of the neural network is not the limiting factor to find a better approximation to the ground state. By performing full-basis simulations, we also eliminated the Monte Carlo sampling noise as another possible suspect, and verified that it does not provide the training bottleneck either.

While incorrect signs constitute an obvious deviation from the exact ground state, by performing separate optimization of the phase and amplitude network, respectively, we found that learning the correct signs is not the sole issue. Also given the sign structure of the exact ground state, our optimization did not result in a more accurate approximation of the wave function amplitudes.

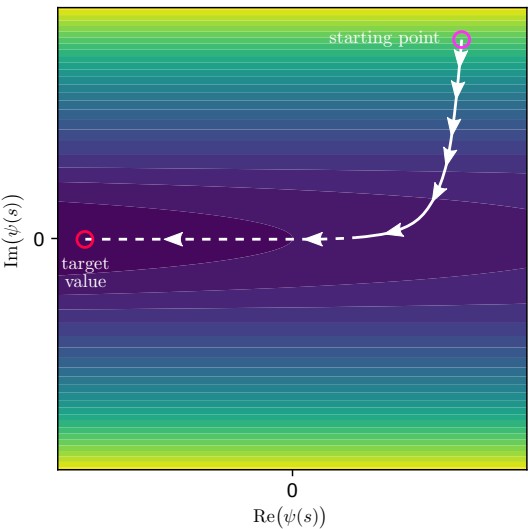

Figure 14: Cartoon of the evolution of a variational wave function coefficient in the complex plane during the optimization. The rapid initial optimization renders the wave function real. Subsequently, reaching the correct sign takes very long, because a narrow saddle has to be passed.

This led us to investigate the properties of the optimization landscape. The numerical experiments we performed suggest that the the $J_1 - J_2$ model has a rugged landscape, consisting of deep valleys abundant in saddles containing just a few highly curved directions. We showed that this topography leads to issues with reproducibility of the simulations. Moreover, scaling up the number of network parameters does not systematically improve the accuracy of the variational energy at $J_2/J_1 = 0.5$, since it is easy for independently initialized simulations to get trapped in one of the many different saddles. Many of these landscape features are shared by spin glasses and, therefore, it is conceivable that the landscape might have a glassy complexity.

## 7.2 Discussion

In combination, the observations from various numerical experiments reported in this work indicate that, when addressing the system size $N = 6 \times 6$, we reach a saddle point in the energy landscape that appears to be very hard to escape. A cartoon of the evolution of individual wave function coefficients is sketched in Fig. 14. During the early stages of optimization, the neural network learns to approximate the Marshall-Peierls sign rule with high accuracy, which renders the wave function real and — together with suited wave function amplitudes — yields already very low energies. To further reduce the energy, the neural network would have to pick up the correct signs also for a small fraction of configurations in the exact ground state that deviate from the Marshall-Peierls sign rule.

Our observation indicates that the route for the neural network to further reduce the energy is to reduce the wave function amplitude of configurations with incorrect sign. Looking at the energy expectation value expressed in terms of the local energy, Eq. (3), it is plausible that this is an efficient way to reduce the energy when the majority of the signs is already produced correctly: for instance, if in $E_\theta^{\mathrm{loc}}(s)$ only the sign of the coefficient $\psi_\theta(s)$ is incorrect, $E_\theta^{\mathrm{loc}}(s)$ will have the wrong sign. Then, the energy obtained by Eq. (3) can be lowered by reducing $p_\theta(s)$. Figure 7 indicates that this is the main mechanism to reduce the energy at later stages of the optimization, because we only find wave function coefficients away from the

main phase peaks at very low amplitudes. We furthermore checked on a number of exemplary configurations that the amplitudes learned by the network are systematically smaller than the exact ground-state result for configurations that deviate from the Marshall-Peierls sign rule.

We conjecture that this required fine tuning of few wave function coefficients constitutes the main obstacle preventing us from reaching a better approximation of the ground state. The spectrum of the Hessian evaluated in the final plateau (Fig. 11) shows that the corresponding saddle reached by the optimization dynamics is largely flat, except for a few eigendirections, which exhibit a strong negative curvature. However, it seems to be very hard for the optimizer to identify these directions — potentially, because the wave function amplitudes of configurations with sign mismatch are strongly suppressed. It is at this point an open question how this issue can be overcome.

In this context, it is worth emphasizing that stochastic reconfiguration is already a second order optimization algorithm that includes awareness of the curvature of the variational manifold. However, it is based on the metric tensor of the neural network manifold and not directly related to the curvature of the energy landscape (the matrix $S$ constitutes only a part of the energy Hessian, see App. G). The sparsity of negative curvature that we revealed through the Hessian spectrum might require a different second order approach. Directly utilizing the Hessian matrix in the optimization algorithm appears straightforward, and, although currently expensive, it is within the scope of present-day computational techniques.

### 7.3 Outlook

It is curious enough to note that for $J_2/J_1 = 0.5$ on the $N = 6 \times 6$ lattice, all works [61, 62, 64], including ours, find comparable variational energies (the energy density difference with the exact one is approximately $2 \cdot 10^{-3}$) while the variational ansatz and optimization procedures are different. This suggests a "universal bottleneck" in the current use of neural networks as variational quantum states for frustrated quantum systems. It remains an open question whether this bottleneck can be overcome or circumvented by other optimization algorithms or enhanced network architectures.

Our results raise the natural question as to why neural quantum states approximate much better the ground states of other models, e.g., Ising, Heisenberg, Bose-Hubbard, etc. [58, 67, 72, 99]. We currently believe that, for these models, even when the variational ansatz is expressive enough to capture the true ground state, independently initialized simulations still end up in distinct landscape minima characterized by different values of the network parameters. However, since these minima all describe the same physical state, they are indistinguishable from the perspective of physics. In more complex setups, such as the $J_2/J_1 = 0.5$ point of the $J_1 - J_2$ model we studied, the degeneracy between the landscape minima is lifted, and a rugged landscape emerges. This makes it particularly hard for optimization algorithms to locate the global minimum (i.e., the lowest-energy configuration).

Unlike tensor network approaches where the quality of the variational approximation is controlled by the amount of entanglement entropy across a cut in the lattice, it is currently unknown what sets the limitations for neural quantum states. While there is no apparent sign problem, "learning" the correct sign structure in the quantum many-body ground state for non-stoquastic Hamiltonians appears in some cases as a nontrivial and challenging computational problem. Figuring out a way to determine the signs would allow one to safely access larger system sizes, and more generally, to deal with generic non-stoquastic Hamiltonians describing frustrated magnets and interacting fermions.

***Note added:*** an alternative study of the complexity of finding the ground state of the $J_1 - J_2$ model using restricted Boltzmann machines can be found in Ref. [108].

## Acknowledgments

We wish to thank E. Altman, Y. Bahri, C. Fisher, and P. Weinberg for valuable discussions, and G. Carleo for helpful comments during peer review. We used JAX [105] for the deep learning implementation and QuSpin [103, 104] for exact diagonalization and model symmetries.

**Funding information**    M.B. was supported by the U.S. Department of Energy, Office of Science, Office of Advanced Scientific Computing Research, under the Accelerated Research in Quantum Computing (ARQC) program, the U.S. Department of Energy under cooperative research agreement DE-SC0009919, the Emergent Phenomena in Quantum Systems initiative of the Gordon and Betty Moore Foundation, and the Bulgarian National Science Fund within National Science Program VIHREN, contract number KP-06-DV-5. M.D. was supported by the U.S. Department of Energy, Office of Science, Office of Basic Energy Sciences, Materials Sciences and Engineering Division under Contract No. DE-AC02-05-CH11231 through the Scientific Discovery through Advanced Computing (SciDAC) program (KC23DAC Topological and Correlated Matter via Tensor Networks and Quantum Monte Carlo). M.S. was supported through the Leopoldina Fellowship Programme of the German National Academy of Sciences Leopoldina (LPDS 2018-07) with additional support from the Simons Foundation. This project has received funding from the European Union's Horizon 2020 research and innovation programme under the Marie Sklodowska-Curie grant agreement No 890711. This research also used resources of the National Energy Research Scientific Computing Center (NERSC), a U.S. Department of Energy Office of Science User Facility operated under Contract No. DE-AC02-05CH11231.

## A    Variational parameters optimization

### A.1    Brief overview of the optimization algorithms

Equations (10) and (12) use gradients $D_k$ to iteratively compute the updates of the neural network parameters $\theta$. Depending on whether we endow the variational manifold with the Euclidean or its intrinsic Riemannian metric, we obtain the expressions,

$$D_k^{\mathrm{E}} = 2\mathrm{Re}\big(F_k\big), \quad D_k^{\mathrm{SR}} = \sum_{k'} (S + \delta \mathbf{1})_{kk'}^{-1} F_{k'},$$

(18)

with $\delta$ an exponentially decaying regularizer needed to prevent instabilities at the early stages of optimization. Formally, $D_k^{\mathrm{E/SR}}$ correspond to the gradients of the energy and the wavefunction overlap cost functions, respectively [69]. Once the gradients are computed, using different optimizers to perform the update will result in various degrees of accuracy. Here, we briefly introduce three common optimizers and compare their performance. More details on gradient descent methods used in machine learning can be found in Ref. [55].

### A.1.1    Stochastic gradient descent

Stochastic Gradient Descent (SGD) is the simplest and most common optimizer, and corresponds to the update rule:

$$\theta_t \longleftarrow \theta_{t-1} - \gamma D_t,$$

(19)

with $\gamma$ denoting the learning rate and $t$ the iteration step. This first-order method is computationally efficient and inexpensive, and can be parallelized easily for speed. The stochastic character of the method in the context of variational quantum Monte Carlo refers to the estimate of the gradients $D$ over the MC sample. The main disadvantages of SGD are that (i) the gradients $D$ of all parameters are weighted by the same learning rate $\gamma$, and (ii) the learning

rate schedule over the iterative optimization procedure is constant in $t$, which can both cause instabilities in the optimization dynamics.

### A.1.2 ADAM

ADAM is another first-order optimizer which estimates a parameter-dependent learning rate from the first and second order moments of the gradients according to [109],

$$
\begin{aligned}
\mathbf{m}_t &\longleftarrow \beta_1 \mathbf{m}_{t-1} + (1-\beta_1)\boldsymbol{D}_t \,, \\
\mathbf{n}_t &\longleftarrow \beta_2 \mathbf{n}_{t-1} + (1-\beta_2)\boldsymbol{D}_t^2 \,, \\
\tilde{\mathbf{m}}_t &\longleftarrow \frac{\mathbf{m}_t}{1-(\beta_1)^t} \,, \\
\tilde{\mathbf{n}}_t &\longleftarrow \frac{\mathbf{n}_t}{1-(\beta_2)^t} \,, \\
\boldsymbol{\theta}_{t+1} &\longleftarrow \boldsymbol{\theta}_t - \gamma \frac{\tilde{\mathbf{m}}_t}{\sqrt{\tilde{\mathbf{n}}_t} + \epsilon} \,.
\end{aligned}
\tag{20}
$$

Besides the learning rate $\gamma$, here $\beta_1 = 0.9$ and $\beta_2 = 0.999$ are the decay rates of the first and second moment estimates, $\epsilon = 10^{-8}$ is a small regularizer, and $(\beta_j)^t$ denotes $\beta_j$ to the power $t$. The main advantage of ADAM is that it can escape narrow and wide valleys of the optimization landscape while retaining computational efficiency which makes it applicable to large neural networks with many independent parameters. Note that a parameter-dependent learning rate can also be achieved by using the SR gradient: although the SR gradients provide a more accurate estimate for the curvature of the parameter manifold, computing the inverse matrix of the curvature matrix $\boldsymbol{S}_{kk'}^{-1}$ of Eq. (12) is expensive for a large number of parameters, and also requires $\boldsymbol{S}_{kk'}$ to be well-conditioned.

### A.1.3 Heun's method

This is a two-stage Runge-Kutta (RK) scheme which allows to adapt the learning rate schedule $\gamma \equiv \gamma_t$ during the optimization process. Depending on the local curvature of the trajectory followed in the parameter manifold, this optimizer can make large steps in flat regions while slowing down in highly-curved regions to efficiently improve the accuracy of the update. In practice, it offers the advantage of not overshooting minima at the later stages of optimization. Additionally, it can stabilize optimization in our setup since large steps correspond to large updates which can implicitly deteriorate the quality of the Monte Carlo sample in subsequent iterations. The major disadvantage of RK methods is their computational cost, since they require multiple evaluations of the gradients per iteration step. We use the implementation described in the supplemental material to Ref. [78] and, in the rest of this paper, we refer to it simply as RK.

Consider the ordinary differential equation $\dot{y} = f(y)$. Setting $y_n = y(t)$, we compute $y_{n+1} = y(t+\tau)$ using

$$
\begin{aligned}
k_1 &= f(y_n) \,, \\
k_2 &= f(y_n + \tau k_1) \,, \\
y_{n+1} &= y_n + \frac{\tau}{2}(k_1 + k_2) \,.
\end{aligned}
\tag{21}
$$

Based on this we can estimate the integration error using varying step sizes $\tau$. If we denote the exact solution by $y(t)$, an integration step with step size $\tau$ yields

$$
y_{n+1} = y(t+\tau) + c\tau^3 \,,
\tag{22}
$$

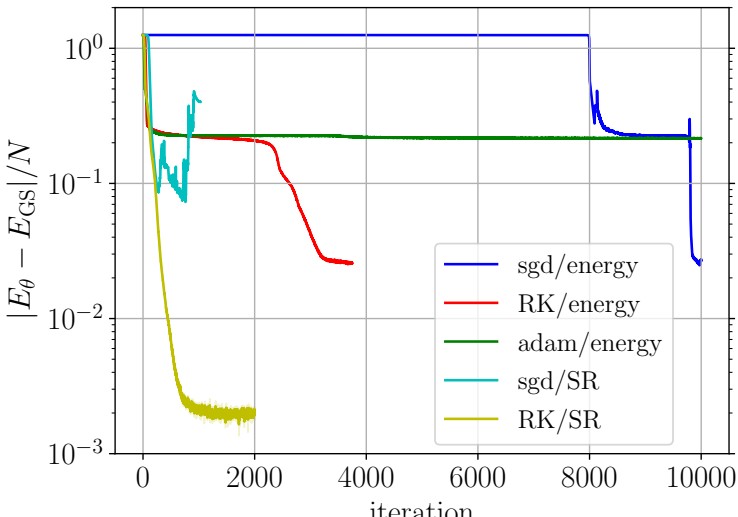

Figure 15: Variational energy density against the iteration number for different optimizers and cost functions. The system size is $N = 6 \times 6$. The network architecture is listed in App. H, and $J_2/J_1 = 0.5$. The MC sample size is $N_{\mathrm{MC}} = 10^5$. The data were produced from simulations using 1024 Haswell cores running for 48 hours.

with an unknown constant $c$, because our integration scheme has an error of $\mathcal{O}(\tau^3)$. Alternatively, we can take two steps of size $\tau/2$, resulting in

$$y'_{n+1} = y(t + \tau) + \underbrace{2c\left(\frac{\tau}{2}\right)^3}_{\delta}, \tag{23}$$

with the integration error $\delta$. The difference of both solutions is

$$\Delta y_{n+1} = ||y_{n+1} - y'_{n+1}|| = \left|\left|\frac{3}{4}c\tau^3\right|\right| = 6||\delta||. \tag{24}$$

Given a desired tolerance $\epsilon$ we can adjust the step size based on this to be

$$\tau' = \tau\left(\frac{\epsilon}{||\delta||}\right)^{1/3}. \tag{25}$$

The choice of a specific norm $||\cdot||$ is in principle arbitrary. Following Ref. [78], we employ the norm induced by the $S$-matrix, $||x||_S = \frac{1}{P}\sqrt{\sum_{k,k'} S_{k,k'} x_k^* x_{k'}}$, for that purpose, meaning that we weigh integration errors by their significance for the physical state. Here, $P$ is the number of variational parameters.

We checked that using a four-stage RK (e.g., Dormand Prince) did not provide any advantage over the cheaper two-stage procedure.

## A.2 Comparison of the optimization algorithms

We now test the behavior of the three optimization algorithms introduced in the previous section on the $J_1 - J_2$ model. We consider both the Euclidean $\boldsymbol{D}_k^{\mathrm{E}}$ and the Riemannian $\boldsymbol{D}_k^{\mathrm{SR}}$ metric gradients (see Sec. 2.1) to perform the updates and display our results in Fig. 15.

We find that SGD (blue curve) is particularly sensitive to the initial condition: the closer the physical state is to an eigenstate of the Hamiltonian, the longer it takes to find the way down

the energy landscape. This is somewhat expected, since it can be shown that all eigenstates are extrema of the energy cost function, where $D^{\mathrm{E}} = 0$. SGD produces a piece-wise constant energy curve which requires a large number of iterations for the variational neural state to get close to the ground state.

In contrast, adaptive learning-rate optimizers, such as ADAM (green curve) and RK (red curve), tend to produce updates that rapidly decrease the energy. For the $J_1 - J_2$ model, we find that ADAM shows a tendency to enter a wide valley of the energy landscape. Escaping this valley appears to be an extremely slow process already for $N = 6 \times 6$ sites due to the rugged character of the landscape, as discussed in Sec. 6. RK, on the other hand, also encounters these wide valleys but succeeds in adapting the learning rate schedule within about 2000 iterations. This allows it to eventually switch to a better valley in the landscape. We emphasize that these results hold in the regime of about $10^3$ neural network parameters, where the optimization problem appears to be constrained.

Unlike the plain energy gradients $D_k^{\mathrm{E}}$, SR takes into account the curvature of the variational manifold by estimating the $S$ matrix from the MC sample to compute the gradients $D_k^{\mathrm{E}}$. Notice that, although SR is designed to find the ground state and thus it effectively performs energy minimization, SR does not formally minimize energy but rather the overlap cost function. Therefore, the SR optimization landscape may in general differ from the energy landscape. This becomes evident for holomorphic neural networks, see Sec. 3.1, where the real part in Eq. (18) induces a difference between the Euclidian and SR gradients even in the flat-metric limit $S \equiv 1$. Bearing a formal similarity to what is commonly known as "Natural Gradient" [69], SR allows to quickly find good directions down the landscape. Intuitively, if we associate gradients with respect to the neural network parameters with directions on the variational manifold, SR weighs them according to their curvature, facilitating the minimization of the overlap cost function.

While it is formally possible to apply all three optimizers in combination with the $D_k^{\mathrm{SR}}$ gradient, we find empirically that ADAM does not converge and we focus on SGD (cyan curve) and RK (yellow curve); this is because ADAM needs a modification to approximate the diagonal elements of the $S$-matrix [69]. For the selected hyperparameters, we find that SGD fails to converge. We believe this is related to an instability associated with the optimal choice of learning rate (we discuss various types of instabilities in Sec. 3). While it is possible to find a network initialization for which the behavior of SGD can be stabilized, we observed that SGD performs worse than RK on average, see Sec. 6 for a discussion of the optimization landscape. In all SR/SGD simulations, we start with $\delta_0 = 100$ and use an exponential decay schedule $\delta_t = \delta_0 e^{-0.075t}$, see Eq. (18). Second-order adaptive RK, on the other hand, combines successfully the benefits of an adaptive learning rate schedule, and the information about the curvature of the variational manifold. Therefore, we select this combination throughout our study. In all SR/RK simulations, we set $\delta = 0$.

Finally, let us make a few comments on the CPU time required by the different combinations of optimizers and cost functions. The energy cost function is much "lighter" than the overlap required for SR because it does not require the computation of the curvature matrix $S$, cf. Eq. (12). Moreover, the absence of $S$ allows one to use much larger neural networks efficiently, since it does not require computing the inverse $S^{-1}$. It is a nontrivial open question whether increasing the number of network parameters can offer any benefits with respect to reaching better energies with first-order energy minimization in the present rugged landscape. Clearly, one downside of using the energy cost function is the number of iterations required to reach a state with a competitive energy, which can well be a few orders of magnitude more, compared to SR.

# B  The convolutional layer

In Sec. 2.2 we introduced the fully-connected (dense) neural network. However, in our simulations, we use the more sophisticated convolutional neural network ansatz. The convolutional layer can be viewed as a specialization of the dense layer, where each output neuron is promoted to a channel,

$$
\boldsymbol{a}_{c,m}^{(l)} = f_l \left( \sum_{c'=1}^{C_{\text{in}}^{(l)}} \sum_{k=1}^{N_F^{(l)}} \tilde{W}_{m,c',k}^{(c,l)} \boldsymbol{a}_{c',\tau_m(k)}^{(l-1)} + \boldsymbol{b}_m^{(l)} \right),
\tag{26}
$$

where the index $c = 1, \ldots, C_{\text{out}}^{(l)}$ introduced in the expression above runs over the number of output channels. The maps $\tau_m(k)$ are chosen to be permutations of the index set corresponding to translations by some fixed stride. Hence, the name convolutional layer. Within each channel, the individual neurons $\boldsymbol{a}_{c,m}^{(l)}$ are coupled with identical weights to the neurons of the previous layer, which were first transformed by the translation $\tau_m(\cdot)$. Therefore, the same kind of information is extracted from all translations.

Moreover, the coupling $\tilde{W}_{m,c',k}^{(c,l)}$ is typically chosen to be sparse, with only $N_F^{(l)}$ non-vanishing elements per input channel, defining a locally constrained receptive field: this means that the convolutional layer can only be sensitive to local features in the input data, see Fig. 1 (b). Nonetheless, global correlations can be captured within this architecture despite the sparsity, by stacking multiple convolutional layers on top of each other. To address all distances, the diameter of the receptive field multiplied by the network depth has to exceed the linear dimension of the input data. Last, note that for stacked convolutional layers, we have by construction $C_{\text{in}}^{(l)} \equiv C_{\text{out}}^{(l-1)}$.

# C  Model symmetries and neural network states

The $J_1 - J_2$ model of Eq. (1) has a number of symmetries that one can take into account to obtain a variational quantum state respecting these important features and to reduce the variational space:

(i) Translation invariance along the two spatial directions $x$ and $y$ (denoted $t_x$ and $t_y$), as we assume periodic boundary conditions,

(ii) The point-group symmetries, which include the reflection (parity) about the $x$-axis, $y$-axis, and the diagonal of the square lattice (denoted $p_x$, $p_y$ and $p_d$),

(iii) Total magnetization conservation,

$$
\left[ \mathcal{H}, S_{\text{tot}}^z = \sum_{j=1}^{N} S_j^z \right] = 0 .
$$

(iv) The spin inversion symmetry when $S_{\text{tot}}^z = 0$, generated by flipping all spins,

(v) A continuous SU(2) spin-rotational symmetry,

$$
\left[ \mathcal{H}, \boldsymbol{S}_{\text{tot}}^2 = \left( \sum_{j=1}^{N} \boldsymbol{S}_j \right)^2 \right] = 0 .
$$

We work with neural networks and optimization procedures which obey symmetries (i–iv), and leave out the continuous SU(2) symmetry. Implementing the SU(2) symmetry in neural quantum states was recently proposed in Ref. [110].

On a finite-size system, the ground state of the $J_1 - J_2$ model is a singlet with zero magnetization which falls in the zero-momentum sector with positive parities and positive spin-inversion quantum numbers. The variational search for the ground state is therefore restricted to these symmetry sectors. First, the zero magnetization sector can be easily enforced via pre-selection by only working with spin configurations $s$ with as many spins pointing up as spins pointing down. Then, the spin inversion symmetry $s_j \mapsto -s_j$ is enacted by considering an even non-linearity activation function $f_{l=1}(\cdot)$ and no biases in the first layer of the neural network.

We considered different approaches to encode the remaining symmetries [66] into the neural quantum state, as discussed in the following.

### C.1 Learning from representatives

In this first approach, one works with representative basis states to incorporate the lattice symmetries (i-ii). The idea is to associate a unique equivalence class under these symmetries to every basis state $s'$, and pick an arbitrary member of the equivalence class $s$ as its representative [1]. This amounts to fixing the symmetry gauge by mapping each spin configuration in the Monte Carlo sample to its representative spin configuration. This way, the network never sees spin configurations other than the representatives.

### C.2 Data symmetrization

Alternatively, one can take a spin configuration $s$, and generate all symmetric spin configurations by applying all possible combinations of symmetries. This augments the original data set by the symmetric configurations. The order in which this is done is irrelevant because all symmetries commute in the ground state sector. This amounts to a total of $L_{t_x} \times L_{t_y} \times 2_{p_x} \times 2_{p_y} \times 2_{p_d} = 8N$ configurations for deep neural networks, or $2_{p_x} \times 2_{p_y} \times 2_{p_d} = 8$ configurations for convolutional neural networks. In convolutional layers, translation symmetries are built into the neural network architecture using translational invariant filters and periodic padding. Note that this procedure results in "double" counting for highly symmetric spin configurations, which map back to themselves before the cyclicity of the symmetry is exhausted. We checked that this double counting does not affect the optimization of neural quantum states. Plus, these highly symmetric spin configurations are exponentially hard to encounter with increasing system size. We then apply the neural network to each one of those symmetry-expanded states separately, and uniformly sum up the final network outputs in the end. The same procedure is carried out for every initial state $s$ which we want to find the amplitude $\psi_\theta(s)$ of. Thus, the neural network becomes invariant under all symmetries used to extend the data set.

### C.3 Relation to symmetrization by quantum number projections

During the preparation of this manuscript another work appeared, that employs quantum number projections in order to symmetrize the variational wave function [80]. In that case, the basis is a variational ansatz $\psi_\theta$ without any symmetrization. For an irreducible representation of the of point group $I$ with character $\chi^I$, momentum quantum numbers $K$ corresponding to the set of translations $T_R$, and spin parity $S_\pm$ the symmetrized wave function is then constructed as

$$\psi_K^{I,S_\pm}(\theta, s) = \sum_{P \in I, R} e^{-i K \cdot R} \chi_P^I \left( \psi_\theta(T_R P \, s) \pm \psi_\theta(T_R P \, s) \right). \tag{27}$$

This symmetrization by quantum number projection differs from our approach in that it symmetrizes at the level of the wave function coefficients, whereas in our approach we symmetrize

the *logarithmic* wave function amplitudes. Thereby, quantum number projections can straight-forwardly address different quantum number sectors, whereas our approach is restricted to momentum $K = 0$ and positive spin parity.

### C.4   Numerical observations

In practice, we observed that data symmetrization is superior to learning from representatives since it allows to systematically reach lower variational energies. While we do not have a formal proof for this, we offer two plausible explanations. Intuitively, this arises due to the loss of locality in the set of representatives: two configurations which differ by an exchange of spins on neighboring sites may have radically different representative spin configurations, and thus it is hard for the neural network to learn the relation between the two. For the same reason, using representatives also defeats the purpose of using convolutional layers. Related to this, representatives are not gauge invariant, and it is currently unclear what the effect of the representative choice (the gauge) on training is. Therefore, all data presented in this paper was generated using the data symmetrization procedure.

We mention in passing yet another empirical observation about the training data: neural networks achieve smaller training errors using the spin configuration convention $s_j \in \{-1, +1\}$, as compared to $s_j \in \{0, 1\}$. This is likely the case since in the $\{0, 1\}$ convention all weights that couple to 0 in the first layer of the network are effectively turned off.

## D   Network initialization

Constructing a deep neural network and optimizing it with Stochastic Reconfiguration may be challenging due to a further kind of instability, caused by inappropriate network initialization. This results in an (almost instantaneous) "blow-up" of the algorithm after a few optimization cycles. This is a manifestation of the so-called vanishing/exploding gradients problem, well-known in the context of machine learning. While the issue is easy to avoid for single-layer networks (via a manual fine-tuning of the distribution which generates the initial weights and biases), it quickly becomes a major problem when experimenting with architectures consisting of several layers.

To overcome this and to enable a seamless initialization of arbitrarily deep networks, we draw the weights and biases $\boldsymbol{\theta}_k^{(l)}$ of each layer $l$ from a uniform distribution on $[-D, +D]$, with a properly adjusted interval size $D$ [78, 97]

$$D = c \Big/ \sqrt{H^{(l)} W^{(l)} \left( C_{\text{in}}^{(l)} + C_{\text{out}}^{(l)} \right)}, \tag{28}$$

for a CNN layer with of dimension $H^{(l)} \times W^{(l)}$ with $C_{\text{in}}^{(l)}$ input channels and $C_{\text{out}}^{(l)}$ output channels. For a DNN where dense layer $l$ has dimensions $n_{\text{in}}^{(l)} \times n_{\text{out}}^{(l)}$, the corresponding expression is obtained as $H^{(1)} W^{(1)} \equiv n_{\text{in}}^{(1)}, C_{\text{in}}^{(1)} = 1, C_{\text{in}}^{(1)} \equiv n_{\text{out}}^{(1)}$ for $l = 1$, and $H^{(l)} W^{(l)} = 1, C_{\text{in}}^{(l)} \equiv n_{\text{in}}^{(l)}, C_{\text{out}}^{(1)} \equiv n_{\text{out}}^{(l)}$ for $l > 1$. This weight normalization ensures that all sums in the forward and backward pass of a neural network with uniformly drawn parameters have unit variance. The constant $c \in [0.1, 10]$ is arbitrary and controls how close the initial state is to the $S^x$-polarized state. In practice we use $c^{\ln} = 0.8$ and $c^{\varphi} = 1$ for the log-amplitude and phase networks, respectively.

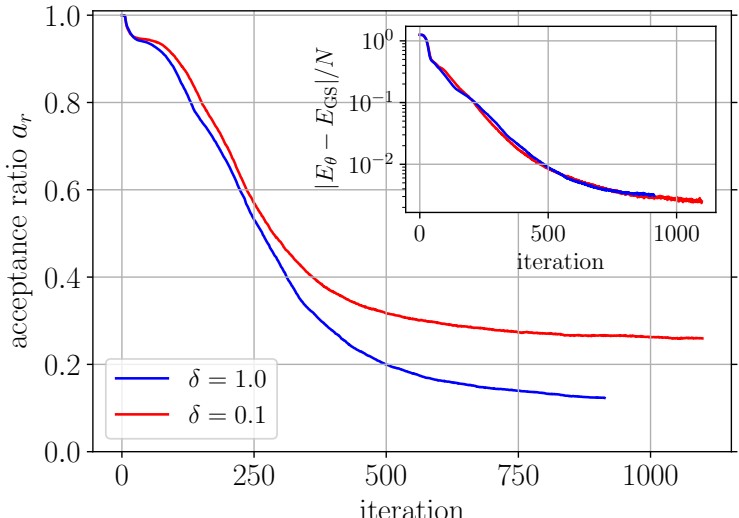

Figure 16: MC acceptance ratio $a_r$ during optimization for global MC proposals and mixed local-global MC proposals for $\delta = 0.1$ (see text). Inset: the corresponding energy training curves. The parameters are the same as in Fig. 7. The system size is $N = 6 \times 6$. The network architecture is listed in App. H, and $J_2/J_1 = 0.5$. Optimization was done using RK in combination with SR, see App. A. The MC sample size is $N_{\text{MC}} = 2^{15}$.

# E   Details of the Monte Carlo simulations

Here, we lay out some details about the Monte Carlo sampling procedure. The goal behind using MC is to find an efficient way to compute expectation values of observables in large Hilbert spaces where using the full basis is infeasible with the present computational power. In this work, we use the Metropolis-Hastings Markov Chain Mote Carlo algorithm to generate a sample of spin configurations from the probability associated with the output of the neural network.

In our MC simulations we consider spin configurations in the zero magnetization sector, which contains the ground state, see Sec. C. To propose the MC updates, we select two lattice sites with opposite spins, and flip them. To select the lattice sites, we used (i) a global scheme where two lattice sites are selected uniformly at random, and (ii) a local scheme where we select a site at random, and exchange its spin with any of its four nearest- and next-nearest neighbors; in both cases we repeat the procedure until the resulting pair of sites has opposite spins. (iii) We can also consider a probabilistic mixture where we apply (i) with probability $\delta$ and (ii) with probability $1 - \delta$.

Each newly proposed configuration $r$ is accepted with probability $p = \min\left(1, |\psi_\theta(r)|^2/|\psi_\theta(s)|^2\right)$ against the current state of the Markov chain $s$. Here, the squared amplitudes are evaluated using the current state of the neural network. To avoid instabilities caused by poor network generalization on the proposed configuration $r$, we additionally require that $|\psi_\theta(r)|^2 \leq \xi|\psi_\theta(s)|^2$ for some number $\xi$ (in practice, we set $\xi = 500$). The acceptance ratio for a given sample is defined as

$$a_r = \#(\text{accepted configurations})\big/N_{\text{MC}}. \tag{29}$$

We set the thermalization (or mixing) time for the MC sampler to $10N$ with $N$ the total number

of lattice sites. The number of proposed updates per sweep, which sets the frequency of storing the state of the Markov chain in the MC sample, is defined in terms of the acceptance ratio of the sample obtained at the previous iteration as $N/2 \times \max(0.05, a_r)$.

In Fig. 7 of the main text, we discussed the phase distribution peaks. A visible difference in the behavior of the $s$-sample and the $s'$-sample at low amplitudes there is the existence of states which do not align with the major phase peaks. Because the $s'$-sample is obtained from the $s$-sample by flipping nearest and next-nearest neighboring spins (see main text), one may wonder if this behavior changes with the use of a local MC update scheme. We checked that a mixed local-global MC proposal scheme with $\delta = 0.1$ does not yield better energies. However it does result in a higher acceptance ratio which speeds up the simulations, see Fig. 16.

Finally, let us mention a potential deadlock for the ground state search using VQMC which we observed a handful of times while experimenting with neural network architectures during the study: if the acceptance ratio falls as low as $a_r = 0.01$, the simulation effectively gets stuck because the time to build the sample becomes too large. In this case, even adaptive learning rate optimizers, such as RK, proved insufficient to escape the deadlock. Hence, it is advisable to restart the simulation with different initial conditions in case such a situation is encountered.

# F   Post-selection of samples

In order for the variational optimization algorithm to work properly, it is crucial to obtain Monte Carlo samples producing reliable estimates of the energy, which can be tricky as one goes down the energy landscape, especially for for $J_2/J_1 = 0.5$. This is because the quality of a batch of $N_{\text{MC}}$ samples $\{s_1, s_2, \cdots s_{N_{\text{MC}}}\}$ depends on the probability distribution $p_\theta(s)$ itself, which is, at the same time, in the process of being optimized.

Since one cannot guarantee that the ground state search dynamics drives the network parameters in regions of parameters space corresponding to easy-to-sample distributions, we adopt a post-selection procedure as follows. Once we have obtained a set of samples at a given iteration step $t$, we evaluate the energy expectation $E_\theta(t) \approx \langle\!\langle E_\theta^{\text{loc}}\rangle\!\rangle$ and its standard deviation $\sigma_{E_\theta}(t) \approx \sqrt{\langle\!\langle |E_\theta^{\text{loc}}|^2\rangle\!\rangle_c}$, according to Eqs. (6) and (7), respectively. The idea is to use these quantities to determine whether to keep the batch of Monte Carlo samples or throw it away and re-sample. We introduce three criteria for re-sampling:

a) $|\text{Re}[E_\theta(t) - E_\theta(t-1)]| < \alpha_1$ ensures that the energy of the variational quantum state is not misestimated. This is particularly useful in cases when the sample at iteration step $t$ estimates a significantly larger value for the energy than at step $t-1$, because the ground state search algorithm is set to minimize, not maximize, the energy.

b) $|\text{Im} E_\theta(t)| < \alpha_2 \sigma_{E_\theta}(t)$ requires that the imaginary part of the sample estimate for the energy is within a given fraction of the energy standard deviation. This reflects an observation that bad samples typically give rise to anomalously large imaginary parts of $E_\theta$ which has to become real-valued in the limit of $N_{\text{MC}} \to +\infty$. Hence, this defines a second natural criterion for re-sampling.

c) $\sigma_{E_\theta}(t) < \alpha_3 \sigma_{E_\theta}(t-1)$ guarantees that the energy standard deviation over the MC sample in two consecutive iteration steps does not increase too rapidly, as expected for a small enough learning rate. A bad MC sample features a large deviation from its mean which can lead to wrong gradient updates. The instabilities we discussed in Sec. 3.1 typically produce large-variance samples, and it is important to be able to detect this behavior. This gives an additional knob to prevent the simulation from blowing up.

In practice, we used $\alpha_1 = 2$, $\alpha_2 = 5$, and $\alpha_3 = 6$. If at iteration $t$ one of the criteria is violated, we go back to iteration $t-1$, load the $(t-1)$-network parameters, and repeat the sampling from there. This eliminates any potentially wrong gradient updates that could have triggered one of the the criteria to fail. We apply sample post-selection only below a certain energy, in our case $E_{\boldsymbol{\theta}} = 0$, which is roughly in the middle of the spectrum, to allow for larger steps in the first iterations.

Last, let us mention that we considered alternative strategies to remedy bad samples, such as discarding outliers falling outside a given quartile as measured by the mean energy. However, this appears to harm the optimization in the early iterations where outliers contain signal about which directions to follow in order to go down the energy landscape.

# G   Energy Hessian matrix

In Sec. 5.2 of the main text, we show data which contains the eigenvalues of the Hessian matrix of the energy cost function. Here, we provide the expression we used to calculate it. For brevity, we only show isolated steps of the derivation.

The starting point is the definition for the energy of the variational state:

$$E_{\boldsymbol{\theta}} = \frac{\langle \psi_{\boldsymbol{\theta}} | \mathcal{H} | \psi_{\boldsymbol{\theta}} \rangle}{\langle \psi_{\boldsymbol{\theta}} | \psi_{\boldsymbol{\theta}} \rangle} = \sum_{\{s\}} p_{\boldsymbol{\theta}}(s) E_{\boldsymbol{\theta}}^{\mathrm{loc}}(s) = \left\langle\!\!\left\langle E_{\boldsymbol{\theta}}^{\mathrm{loc}} \right\rangle\!\!\right\rangle, \tag{30}$$

where we used $\langle\!\langle \cdot \rangle\!\rangle = \sum_{\{s\}} p_{\boldsymbol{\theta}}(s)(\cdot)$. Taking the derivative with respect to the variational parameters $\boldsymbol{\theta}$, the energy gradient $\nabla_{\boldsymbol{\theta}_n} \equiv \partial_n$ amounts to

$$\partial_n E_{\boldsymbol{\theta}} = 2\mathrm{Re} \left\langle\!\!\left\langle O_n^* \Delta E_{\boldsymbol{\theta}}^{\mathrm{loc}} \right\rangle\!\!\right\rangle, \tag{31}$$

with the short-hand definitions $O_n(s) = \partial_n \ln \psi_{\boldsymbol{\theta}}(s)$, and $\Delta E_{\boldsymbol{\theta}}^{\mathrm{loc}}(s) = E_{\boldsymbol{\theta}}^{\mathrm{loc}}(s) - \langle\!\langle E_{\boldsymbol{\theta}}^{\mathrm{loc}} \rangle\!\rangle$. Taking into account the $\boldsymbol{\theta}$ dependence in $p_{\boldsymbol{\theta}}(s)$, $O_n^*(s)$, $\Delta E_{\boldsymbol{\theta}}^{\mathrm{loc}}(s)$, the entries of the Hessian matrix can be computed as

$$
\begin{aligned}
H_{mn} = \partial_m \partial_n E_{\boldsymbol{\theta}} = \quad & 4 \left\langle\!\!\left\langle \mathrm{Re}\,(O_m) \mathrm{Re}\left(O_n^* \Delta E_{\boldsymbol{\theta}}^{\mathrm{loc}}\right) \right\rangle\!\!\right\rangle \\
- \quad & 4\left[ \left\langle\!\!\left\langle \mathrm{Re}\,(O_m) \right\rangle\!\!\right\rangle \left\langle\!\!\left\langle \mathrm{Re}\left(O_n^* \Delta E_{\boldsymbol{\theta}}^{\mathrm{loc}}\right) \right\rangle\!\!\right\rangle + (m \leftrightarrow n) \right] \\
+ \quad & 2 \left\langle\!\!\left\langle \mathrm{Re}\left(\partial_m \left[O_n^*\right] \Delta E_{\boldsymbol{\theta}}^{\mathrm{loc}}\right) \right\rangle\!\!\right\rangle \\
+ \quad & 2\mathrm{Re}\left( \sum_{\{s\}} p_{\boldsymbol{\theta}}(s) O_n^*(s) \times \frac{1}{\psi_{\boldsymbol{\theta}}(s)} \sum_{\{s'\}} \mathcal{H}_{ss'} \psi_{\boldsymbol{\theta}}(s') O_m(s') \right) \\
- \quad & 2 \left\langle\!\!\left\langle \mathrm{Re}\left(O_n^* O_m E_{\boldsymbol{\theta}}^{\mathrm{loc}}\right) \right\rangle\!\!\right\rangle. \tag{32}
\end{aligned}
$$

For a variational ansatz with two real-valued neural networks (one for the phase, and the other for the log-probability amplitude of the wavefunction), see Sec. 3, the expression for the Hessian takes the form

$$
\begin{aligned}
H_{mn} = \partial_m \partial_n E_{\boldsymbol{\theta}} = \quad & 2 \left\langle\!\!\left\langle \partial_m O_n^{\ln} \mathrm{Re}(\Delta E_{\boldsymbol{\theta}}^{\mathrm{loc}}) + \partial_m O_n^{\varphi} \mathrm{Im}(\Delta E_{\boldsymbol{\theta}}^{\mathrm{loc}}) \right\rangle\!\!\right\rangle \\
+ \quad & 2 \sum_{\{s\}} p_{\boldsymbol{\theta}}(s) \left[ O_n^{\ln}(s) \mathrm{Re}\left( \frac{1}{\psi_{\boldsymbol{\theta}}(s)} \sum_{\{s'\}} \mathcal{H}_{ss'} \psi_{\boldsymbol{\theta}}(s') O_m(s') \right) \right] \\
+ \quad & 2 \sum_{\{s\}} p_{\boldsymbol{\theta}}(s) \left[ O_n^{\varphi}(s) \mathrm{Im}\left( \frac{1}{\psi_{\boldsymbol{\theta}}(s)} \sum_{\{s'\}} \mathcal{H}_{ss'} \psi_{\boldsymbol{\theta}}(s') O_m(s') \right) \right] \\
- \quad & 4 \left[ \left\langle\!\!\left\langle O_m^{\ln} \right\rangle\!\!\right\rangle \left\langle\!\!\left\langle O_n^{\ln} \mathrm{Re}(\Delta E_{\boldsymbol{\theta}}^{\mathrm{loc}}) + O_n^{\varphi} \mathrm{Im}(\Delta E_{\boldsymbol{\theta}}^{\mathrm{loc}}) \right\rangle\!\!\right\rangle + (m \leftrightarrow n) \right] \\
+ \quad & 2 \left\langle\!\!\left\langle \left(O_m^{\ln} O_n^{\ln} - O_m^{\varphi} O_n^{\varphi}\right) \mathrm{Re}(\Delta E_{\boldsymbol{\theta}}^{\mathrm{loc}}) \right\rangle\!\!\right\rangle \\
+ \quad & 2 \left\langle\!\!\left\langle \left(O_m^{\ln} O_n^{\phi} + (m \leftrightarrow n)\right) \mathrm{Im}(\Delta E_{\boldsymbol{\theta}}^{\mathrm{loc}}) \right\rangle\!\!\right\rangle \\
- \quad & 2 \left\langle\!\!\left\langle O_m^{\ln} O_n^{\ln} + O_m^{\varphi} O_n^{\varphi} \right\rangle\!\!\right\rangle \left\langle\!\!\left\langle \mathrm{Re}(E_{\boldsymbol{\theta}}^{\mathrm{loc}}) \right\rangle\!\!\right\rangle. \tag{33}
\end{aligned}
$$

Here, we used $O_n = O_n^{\ln} + iO_n^{\varphi}$, where $O_n^{\ln}(\boldsymbol{s}) = \partial_n \ln|\psi_{\boldsymbol{\theta}}(\boldsymbol{s})|$, and $O_n^{\varphi}(\boldsymbol{s}) = \partial_n \varphi_{\boldsymbol{\theta}}(\boldsymbol{s})$[9]. Notice that the Hessian matrix is *not* block-diagonal in the log-amplitude and phase networks parameter spaces, i.e., it couples the parameters of the log-amplitude and phase networks. We also see that the $\boldsymbol{S}$-matrices for the log-amplitude and phase networks enter in the last line of the Hessian. Finally, one can readily convince oneself that the above expression defines a symmetric matrix, as is expected for the Hessian due to the commutativity of the two partial derivatives.

## H Neural network architectures

Table 1: Neural network architectures used for the numerical simulations shown throughout the paper.

| Figure | Layer 1 | Layer 2 | Layer 3 | Layer 4 | Layer 5 | Layer 6 |
|---|---|---|---|---|---|---|
| Fig. 2 (4 × 4) holomorphic 296 cpx. parameters (592 real parameters) | Dense 12 neurons | Dense 8 neurons | — | — | — | — |
| Fig. 2 (6 × 6) holomorphic 536 cpx. parameters (1072 real parameters) | Dense 12 neurons | Dense 8 neurons | — | — | — | — |
| Fig. 3 holomorphic 432 cpx. parameters (864 real parameters) | Dense 12 neurons | — | — | — | — | — |
| Fig. 6 log-amp. net 521 real parameters | Convolutional 6 6 × 6 filters | Convolutional 4 3 × 3 filters | Convolutional 4 2 × 2 filters | Dense 2 neurons | Dense 2 neurons | Regularizer |
| Fig. 6 phase-net 572 real parameters | Dense 12 neurons | Dense 8 neurons | Dense 4 neurons | — | — | — |
| Fig. 6 log-amp. net 521 real parameters | Convolutional 7 6 × 6 filters | Convolutional 5 3 × 3 filters | Convolutional 3 2 × 2 filters | Dense 2 neurons | Dense 2 neurons | Regularizer |
| Fig. 6 phase-net 720 real parameters | Dense 14 neurons | Dense 10 neurons | Dense 6 neurons | — | — | — |
| Fig. 6 log-amp. net 857 real parameters | Convolutional 8 6 × 6 filters | Convolutional 6 3 × 3 filters | Convolutional 4 2 × 2 filters | Dense 4 neurons | Dense 2 neurons | Regularizer |
| Fig. 6 phase-net 884 real parameters | Dense 16 neurons | Dense 12 neurons | Dense 8 neurons | — | — | — |

---

[9]Note that in the second and third lines of Eq. (33), it is the full $O_m(\boldsymbol{s}')$ that enters the sum over $\{\boldsymbol{s}'\}$, not its phase or log component.

| Figure | Layer 1 | Layer 2 | Layer 3 | Layer 4 | Layer 5 | Layer 6 |
|---|---|---|---|---|---|---|
| Fig. 6<br>log-amp. net<br>1325 real parameters | Convolutional<br>10 $6 \times 6$ filters | Convolutional<br>8 $3 \times 3$ filters | Convolutional<br>6 $2 \times 2$ filters | Dense<br>4 neurons | Dense<br>2 neurons | Regularizer |
| Fig. 6<br>phase-net<br>1260 real parameters | Dense<br>20 neurons | Dense<br>16 neurons | Dense<br>12 neurons | — | — | — |
| Fig. 7<br>log-amp. net<br>857 real parameters | Convolutional<br>8 $6 \times 6$ filters | Convolutional<br>6 $3 \times 3$ filters | Convolutional<br>4 $2 \times 2$ filters | Dense<br>4 neurons | Dense<br>2 neurons | Regularizer |
| Fig. 7<br>phase-net<br>884 real parameters | Dense<br>16 neurons | Dense<br>12 neurons | Dense<br>8 neurons | — | — | — |
| Fig. 8 ($4 \times 4$)<br>log-amp. net<br>697 real parameters | Convolutional<br>8 $6 \times 6$ filters | Convolutional<br>6 $3 \times 3$ filters | Convolutional<br>4 $2 \times 2$ filters | Dense<br>4 neurons | Dense<br>2 neurons | Regularizer |
| Fig. 8 ($4 \times 4$)<br>phase-net<br>884 real parameters | Dense<br>16 neurons | Dense<br>12 neurons | Dense<br>8 neurons | — | — | — |
| Fig. 8 ($6 \times 6$)<br>log-amp. net<br>857 real parameters | Convolutional<br>8 $6 \times 6$ filters | Convolutional<br>6 $3 \times 3$ filters | Convolutional<br>4 $2 \times 2$ filters | Dense<br>4 neurons | Dense<br>2 neurons | Regularizer |
| Fig. 8 ($6 \times 6$)<br>phase-net<br>884 real parameters | Dense<br>16 neurons | Dense<br>12 neurons | Dense<br>8 neurons | — | — | — |
| Fig. 10<br>log-amp. net<br>857 real parameters | Convolutional<br>8 $6 \times 6$ filters | Convolutional<br>6 $3 \times 3$ filters | Convolutional<br>4 $2 \times 2$ filters | Dense<br>4 neurons | Dense<br>2 neurons | Regularizer |
| Fig. 10<br>phase-net<br>884 real parameters | Dense<br>16 neurons | Dense<br>12 neurons | Dense<br>8 neurons | — | — | — |
| Fig. 12<br>log-amp. net<br>857 real parameters | Convolutional<br>8 $6 \times 6$ filters | Convolutional<br>6 $3 \times 3$ filters | Convolutional<br>4 $2 \times 2$ filters | Dense<br>4 neurons | Dense<br>2 neurons | Regularizer |
| Fig. 12<br>phase-net<br>884 real parameters | Dense<br>16 neurons | Dense<br>12 neurons | Dense<br>8 neurons | — | — | — |
| Fig. 9<br>log-amp. net<br>857 real parameters | Convolutional<br>8 $6 \times 6$ filters | Convolutional<br>6 $3 \times 3$ filters | Convolutional<br>4 $2 \times 2$ filters | Dense<br>4 neurons | Dense<br>2 neurons | Regularizer |
| Fig. 9<br>phase-net<br>884 real parameters | Dense<br>16 neurons | Dense<br>12 neurons | Dense<br>8 neurons | — | — | — |
| Fig. 13 ($4 \times 4$)<br>holomorphic<br>64 cpx. parameters<br>(128 real parameters) | Dense<br>4 neurons | — | — | — | — | — |
| Fig. 15<br>log-amp. net<br>857 real parameters | Convolutional<br>8 $6 \times 6$ filters | Convolutional<br>6 $3 \times 3$ filters | Convolutional<br>4 $2 \times 2$ filters | Dense<br>4 neurons | Dense<br>2 neurons | Regularizer |
| Fig. 15<br>phase-net<br>884 real parameters | Dense<br>16 neurons | Dense<br>12 neurons | Dense<br>8 neurons | — | — | — |
| Fig. 16<br>log-amp. net<br>857 real parameters | Convolutional<br>8 $6 \times 6$ filters | Convolutional<br>6 $3 \times 3$ filters | Convolutional<br>4 $2 \times 2$ filters | Dense<br>4 neurons | Dense<br>2 neurons | Regularizer |
| Fig. 16<br>phase-net<br>884 real parameters | Dense<br>16 neurons | Dense<br>12 neurons | Dense<br>8 neurons | — | — | — |

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
