# Peer review of "Learning the ground state of a non-stoquastic quantum Hamiltonian in a rugged neural network landscape"

_SciPost Physics, doi:SciPost Phys. 10, 147 (2021)_

## Round 1 · Referee Report · Anonymous (Referee 1) · 2021-1-16

Strengths

1- discuss important issues on neural-network wave functions, which are often used to study strongly-correlated systems.

Weaknesses

1- the presentation is confused and it is not easy to follow the logic of the presentation.

Report

In this work Bukov and collaborators assess the accuracy of neural-network wave functions for the frustrated J1-J2 model on the square lattice.

I think that the paper is quite hard to read and thread of the conversation is not very linear. The general remark is that several different problems are shown, but the reader may be lost in all details.

I will try to make some suggestion to improve the presentation, possibly cutting/rearranging the discussion.

1) Section 3: I do not really understand why the discussion of the complex (holomorphic) state is so long, since, at the end of the day, it does not give sensible results. Still, I do not really understand the reason why there are numerical instabilities. The fact that a complex function must diverge somewhere in the complex plan does not convince me too much. Anyway, since the parametrization with two real functions is eventually more stable, I suggest to cut the part on the complex wave function, just showing the most important results.

2) Section 4: This section looks useful, even though I would have preferred something a bit more quantitative. For the case they show, the exact sign is almost given by the Marshall rule, but what is the relevance of configurations violating the Marshall sign? Namely, what is the amplitude of these configurations (exact vs variational ones)? I am not sure about the real lesson that we learn from this section.

3) Sections 5 and 6: I think that this is most confusign part. For example, I would have discussed the "full basis simulation" in tight connection with "Varying the size of the Monte Carlo fluctuations", since I think they are directly related.

My main concern is about the fact that the machine precision may really affect the results. In my experience, this suggests that there is some problem in the numerical code. Also the fact that different random number sequences may affect the results is very suspicious. There is no a very convincing discussion about that.

4) Section 7: I would just shorten the discussion.

Now, let me give a general comment: all the paper is about small lattices (e.g., 4x4 and 6x6), for which exact diagonalizations are possible. What is the usefulness of these neural-network states if 1) they provide more or less the same accuracy of much simpler states (e.g., Gutzwiller-projected fermionic states) and 2) calculations are limited to small systems?

A minor point: it would be better to specify the value of J_2/J_ 1 along the text and in the captions of the figures.

In summary, I would strongly suggest a deep revision of the paper, which should make the presentation linear.

Requested changes

See report.

  • validity: ok
  • significance: good
  • originality: ok
  • clarity: poor
  • formatting: reasonable
  • grammar: excellent

Author:  Marin Bukov  on 2021-04-13  [id 1356]

(in reply to Report 1 on 2021-01-16)

We would like to thank the Referee for the insightful report. Before we answer in detail to all points raised, let us make a few general statements about the scope of the paper:

Neural quantum states are a new field of computational physics, which emerged only in the last 3-4 years. While using neural networks as variational quantum states is tempting, because they represent universal approximators, one has to apply the method with care.

Unlike tensor networks (based on quantum entanglement), it is currently unknown how to universally control the quality of the approximation in neural networks, irrespective of the physical system studied. In the literature, one often encounters the typical "just add more neurons and all will be good", but for complex problems, such as the J1-J2 model under consideration, this is not as straightforward. Hence, the field is currently in its initial rapid development stage, and we believe it is important at this stage to also pay attention to some technical aspects of the method, without necessarily producing an outstanding new insight into the physics of 2D lattice systems. Such insights are the ultimate goal and they will inevitably follow once the inner workings of the method are understood.

The present manuscript is thus intended to discuss shortcomings of the method, and to point out those parts of it which have to be handled with particular care. These are best exemplified using the J1-J2 model at J2/J1=0.5. In that sense, we restrict the discussion to the smallest system size, where typical issues occur and where exact validation of the results is possible, and we have clearly stated this in the introduction. For these reasons, part of the discussion is devoted to widely-used paradigms, such as holomorphic neural networks, or the occurrence of training instabilities of various kinds: this is deliberate and its purpose is to focus the attention of the expert community. This also motivates the explorative stance of the text, with a number of numerical experiments suggested and performed, that shed light onto the same problem but from a different perspective, in the attempt to build a more complete picture. Therefore, in writing the paper, we have paid specific attention to separate plausible explanations and conclusions that can be drawn from the data, from immediate facts [see Sec 7]. We believe that this is crucial at this stage of development in the field, since one can encounter non-reproducible results and erroneous or superficial claims in the literature.

1)

  • Section 3: I do not really understand why the discussion of the complex (holomorphic) state is so long, since, at the end of the day, it does not give sensible results. [...] Anyway, since the parametrization with two real functions is eventually more stable, I suggest to cut the part on the complex wave function, just showing the most important results.

The holomorphic approximation is currently widely applied in many studies in the field. As we pointed out above, we believe it is particularly important for the readers from the community to appreciate some of the (theoretical) deficiencies here. We refrain from commenting on whether it produces "sensible results", since the field is undergoing rapid development and it is important to keep an open mind about all available techniques, instead of discarding them right away [see, e.g., Ref https://iopscience.iop.org/article/10.1088/1361-648X/abe268/meta mentioned by Referee 1]. Therefore, we prefer to keep this section in the text.

  • Still, I do not really understand the reason why there are numerical instabilities. The fact that a complex function must diverge somewhere in the complex plan does not convince me too much.

We added a new figure to this section, and a corresponding paragraph to explain the mechanism in detail. The essence is, that encountering a pole on any of the nodes/neurons of the neural network for a single arbitrary input configuration, is sufficient to cause the instability. The large sample size and the large number of nodes/neurons in the network ansatz renders it almost certain to encounter this problem.

2)

  • Section 4: This section looks useful, even though I would have preferred something a bit more quantitative. For the case they show, the exact sign is almost given by the Marshall rule, but what is the relevance of configurations violating the Marshall sign?

Note that configurations violating the Marshall-Peierls sign rule are those that prevent the algorithm from lowering the energy further. In theory, these configurations are expected to contribute significantly to the gradient update, so that the network parameters can be tuned in a direction that lowers the energy. In practice, however, they do not necessarily occur in the s-sample (though some of them may show up in the s'-sample), and thus their contribution to the energy is suppressed.

  • Namely, what is the amplitude of these configurations (exact vs variational ones)?

Take the exact ground state: 50.22% of all spin configurations have a sign, different from the sign predicted by Marshall-Peierls rule. At first sight, this suggests that it should not be too difficult to learn an approximation to the exact sign distribution. However, all these configurations (the 50.22%) taken together constitute only 1.95% of the norm of the exact ground state. This tells us that the Marshall-violating configurations have tiny amplitudes in the exact ground state, and hence they are unlikely to be encountered in MC sampling. This provides a plausible explanation for why the Marshall-Peierls sign rule occurs as an attractor, i.e. independent studies all report that neural networks tend to learn the Marshall-Peierls signs, rather than the exact GS sign distribution.

In the variational state, the neural network parameters are adjusted during optimization to suppress the amplitudes of these configurations, compared to their exact values. In this way, the value of the empirical (i.e. MC-sampled) loss can be kept minimal; however, the resulting sign distribution remains far from the exact GS sign distribution.

We replaced the last subfigure in Fig 7 to make this comparison obvious, and added a corresponding discussion to the text.

  • I am not sure about the real lesson that we learn from this section.

We have re-written the section. In particular, (i) we now make it clear in the introductory paragraph what we do and why we do it; (ii) we moved the paragraph on the skewedness of the distributions in Fig 6 to a footnote; (iii) we re-structured the discussion of the s- and s'-samples to make it clearer; (iv) we added a conclusion paragraph with a clear message to summarize this short section of the paper.

3)

  • Sections 5 and 6: I think that this is most confusing part. For example, I would have discussed the "full basis simulation" in tight connection with "Varying the size of the Monte Carlo fluctuations", since I think they are directly related.

We agree with the Referee's suggestion and we moved the figure on the size of MC fluctuations into "Full basis simulation" section.

  • My main concern is about the fact that the machine precision may really affect the results. In my experience, this suggests that there is some problem in the numerical code.

Typically, when one has doubts about the numerical code having problems, the best check to do is to write a new code from scratch (ideally written by a different person), and compare the results. Indeed, we have tested two completely independently written codes back to back (one in C++ and the other -- in Python): note that the difference between the two codes is shown in the inset to Fig 12. For up to 150 iterations of the algorithm, the difference between the two curves is at machine precision level. We believe this certifies the correctness of both codes.

  • Also the fact that different random number sequences may affect the results is very suspicious. There is no a very convincing discussion about that.

We believe that this fact is not as surprising, given the picture of a highly rugged variational landscape that we propose. Neural network landscapes have tons of parameters and thus host many local minima and saddles in the parameter landscape (similar to spin glasses). Using a different seed for the random number generator means that the training data (sampled using MC) will vary, and so will the values of the gradient updates. Thus, two optimization trajectories on the variational manifold will naturally diverge, and eventually they will get stuck in different local minima/saddles, and this is precisely the behavior we observe.

On a more general level, numerical precision can have drastic effects in computational methods. For instance, in tensor networks simulations, if a symmetry such as U(1) is not explicitly encoded, the state will eventually lose it during the optimization process while it was (i) initially symmetric up to machine precision and (ii) the optimization dynamics should not theoretically affect this. This behavior occurs as a result of limited machine precision.

4)

  • Section 7: I would just shorten the discussion.

In Section 7 we summarize and discuss the various findings that we presented throughout the paper. We believe that these deliberations are important for the reader's comprehension of our multifaceted study. Therefore, we refrained from shortening it. Instead, we introduced a pictorial illustration of the optimization process as we infer it from our observations in Fig. 14, which should render our discussion better accessible for the reader.

We emphasize once more that we deliberately separate the summary of results from the discussion in our conclusions in order to clearly distinguish facts from interpretation.

  • Now, let me give a general comment: all the paper is about small lattices (e.g., 4x4 and 6x6), for which exact diagonalizations are possible. What is the usefulness of these neural-network states if 1) they provide more or less the same accuracy of much simpler states (e.g., Gutzwiller-projected fermionic states) and 2) calculations are limited to small systems?

As we discussed in the preliminary comments, it is not the purpose of this paper to scale up the simulation to big system sizes. This has been done on the J1-J2 model, but it produces unsatisfactory results: even though the relative difference between the variational energy and the exact ground state energy can be reduced to as low as 10^{-3}, the variational state is still far away from the ground state in terms of its physical properties. This raises the general question as to how to measure the accuracy of the approximation. Hence, it is more appropriate to study small system sizes, where a direct comparison to the exact ground state is possible. We have detailed our motivation as to why we do not show results on larger systems -- namely, we want to validate the variational state with the exact results -- at the beginning of Sec 1.2 in the introduction, and then at the end of Sec 2.1 of the paper.

Regarding the "usefulness" of neural quantum states, note that neural network states can easily reach large system sizes for other Hamiltonians, so we prefer to stay open-minded and not discard them as useless at this stage.

  • A minor point: it would be better to specify the value of J_2/J_ 1 along the text and in the captions of the figures.

We implemented the suggestion.

---

## Round 1 · Referee Report · Giuseppe Carleo (Referee 2) · 2021-2-9

Strengths

Clear and detailed account of numerical experiments
Innovative techniques to improve optimization of neural quantum states
Insight in numerical instability

Weaknesses

None to report

Report

This work is a very interesting study of frustrated models with neural quantum states, and highlights some very important technical aspects and challenges of these variational studies.

The paper is very well written and it is a trove of precious technical details and novel parameterization strategies that will help the community mover forward in this fast-developing field.

I do not have any "blocking" remarks and I recommend to publish the paper, essentially, as it is. I just have a few optional remarks the authors might want to address:

  1. Recent works (starting from https://journals.aps.org/prresearch/abstract/10.1103/PhysRevResearch.2.033075 ) have shown that separately optimizing the sign and the amplitudes is beneficial in getting robust results (see also this recent work with an alternated optimization strategy https://arxiv.org/abs/2101.08787 ). In this regard, I find the "partial learning" experiments done in this work quite interesting but also hard to understand (at least , for me!). What is the neural network architecture/ size used in the partial learning? Maybe it is written in the text but I could not find it. If the exact sign structure is taken, is there a significative dependence on the network size?

  2. Something I believe it would be nice to discuss more is the role of symmetries in improving the optimization landscape. For example, it is quite remarkable that a very simple, shallow RBM (purely holomorphic!) has obtained significantly improved results over the CNN (and DMRG) at the infamous J_2/J_1=0.5 https://iopscience.iop.org/article/10.1088/1361-648X/abe268/meta . Do the authors have insight on why this might be the case, in light also of their more general understanding?

  3. A very small remark I also already privately mailed to the authors is about the remark in the conclusion "Directly utilizing the Hessian matrix appears straightforward, but it is, unfortunately, prohibitively expensive.". I am not sure about these conclusions, because one can use the same kind of tricks that are used to apply the quantum geometric tensor (S matrix) on the gradients, without forming the actual matrix. So I believe it should be possible to have a relatively decent scaling also for the Hessian. I would maybe leave this a bit more open as it is currently written.

  • validity: high
  • significance: high
  • originality: good
  • clarity: high
  • formatting: excellent
  • grammar: excellent

Author:  Marin Bukov  on 2021-04-13  [id 1355]

(in reply to Report 2 by Giuseppe Carleo on 2021-02-09)

1) alternating training

  • What is the neural network architecture/size used in the partial learning? Maybe it is written in the text but I could not find it.

For the log-amplitude network, we use a 6-layer neural network: three convolutional layers followed by two fully-connected layers plus a regularizer [697 parameters in total]; we use three fully connected layers for the phase network [884 parameters in total]. This network architecture is the same architecture as in the numerical simulations without partial learning, and is listed in the table of App. H.

  • If the exact sign structure is taken, is there a significative dependence on the network size?

We did not perform partial learning simulations with the exact sign structure, since our motivation was to test the dependence on the Monte-Carlo noise. It would be interesting to investigate this in future work.

With regard to the alternated optimizations strategy, we did explore a few variants, including updating only one of the two (phase and log-amplitude) nets for a fixed number M of training iterations (say we repeat training the log-amplitude network for M=5 iterations, then switch to the phase net for M=5 more iterations, and so on). We also tried the straightforward alternating training approach where each iteration we flip the network to be optimized (M=1). We did not see an immediate advantage in terms of reaching a lower energy, but we also did not do an extensive hyperparameter search. One technical advantage of this alternating training is that one can recycle the sample from the last log-amplitude network training iteration and re-use it during the entire phase network training stage, because the probability distribution remains unchanged when the phase net is optimized.

2)

  • Something I believe it would be nice to discuss more is the role of symmetries in improving the optimization landscape. For example, it is quite remarkable that a very simple, shallow RBM (purely holomorphic!) has obtained significantly improved results over the CNN (and DMRG) at the infamous J_2/J_1=0.5 https://iopscience.iop.org/article/10.1088/1361-648X/abe268/meta . Do the authors have insight on why this might be the case, in light also of their more general understanding?

Indeed, Ref https://iopscience.iop.org/article/10.1088/1361-648X/abe268/meta, obtains a lower energy with a single-layer RBM, but note that their ansatz assumes the Marshall-Peierls sign rule. We also note that their network is in fact "partially holomorphic" with some real and some complex couplings; moreover, the network sizes used are rather large (up to 14k real parameters). Our observation is that simple holomorphic RBM's appear to struggle to learn without adopting the Marshall-Peierls sign rule (see Sec. III A). We thank the Referee for pointing out this relevant reference to us, and we included it in the paper.

Regarding the symmetries, we added a brief discussion of how their method differs from using representatives and input-symmetrization in App. C.

3)

  • A very small remark I also already privately mailed to the authors is about the remark in the conclusion "Directly utilizing the Hessian matrix appears straightforward, but it is, unfortunately, prohibitively expensive.". I am not sure about these conclusions, because one can use the same kind of tricks that are used to apply the quantum geometric tensor (S matrix) on the gradients, without forming the actual matrix. So I believe it should be possible to have a relatively decent scaling also for the Hessian. I would maybe leave this a bit more open as it is currently written.

Indeed, one has to distinguish between having an efficient optimizer using the Hessian instead of the Fisher matrix (which, as the Referee points out, is feasible using modern parallelization techniques and GPUs), and obtaining the spectrum of the exact Hessian, which we believe is still beyond the reach of present-day computational techniques for neural networks with more than few thousand parameters. We thank the Referee for this comment, and we modified the text to avoid any confusion.

---

## Round 2 · Referee Report · Anonymous · 2021-5-9

Report
I have read the new version of the manuscript and the responses to my previous questions. I think that the presentation has been improved, even though it is not optimal. I still have a few points that are not completely clear:
1) According to the text (page 10), the Fig.2 has been obtained with a single layer NN. However, in App.H, it seems that the NN has 2 layers. Am I wrong?
2) I am not sure to understand the whole construction of Fig.4: why are we looking to z_i^(1)? By the way, how many layers are there? In addition, I do not understand the location of the poles in Fig.4: is there a way to obtain them?
3) I do not fully understand the discussion in Sec.5: according to the results of Sec.4, the 50% of the configurations not having the Marshall sign rule have a tiny contribution to the ground-state wave function. It should be possible that this is the best that can be done with the chosen number of parameters and, in order to improve the correct signs, a larger number of parameters is necessary. Could the author comment on that?
In summary, I think that the paper is interesting and could be published, after these additional points will be addressed.
Author: Marin Bukov on 2021-05-31 [id 1480]
(in reply to Report 1 on 2021-05-09)1) According to the text (page 10), the Fig.2 has been obtained with a single layer NN. However, in App.H, it seems that the NN has 2 layers. Am I wrong?
2) I am not sure to understand the whole construction of Fig.4: why are we looking to z_i^(1)?
By the way, how many layers are there?
In addition, I do not understand the location of the poles in Fig4: is there a way to obtain them?
3) I do not fully understand the discussion in Sec.5: according to the results of Sec.4, the 50% of the configurations not having the Marshall sign rule have a tiny contribution to the ground-state wave function.
It should be possible that this is the best that can be done with the chosen number of parameters and, in order to improve the correct signs, a larger number of parameters is necessary. Could the author comment on that?
On a separate note, notice that the number of network parameters cannot be enlarged indefinitely; from a given critical value onwards (about several thousand, depending on the computing architecture) computing the S-matrix becomes infeasible.

---

## Round 2 · Referee Report · Giuseppe Carleo · 2021-5-29

Report
I thank the authors for having addressed my comments, I feel that the work is now publishable as it is.

---

## Round 2 · Author Response

thank you for considering our manuscript for review. We have revised the manuscript taking into consideration all critique points raised by the referees.
We hope that with the revisions made, our work meets the publication criteria of SciPost.
Sincerely,
the authors

---

## Round 2 · List of Changes

- clarified sentences throughout the text
- added new Fig 4 to explain the occurrence of a potential instability with holomorphic neural networks, and a corresponding paragraph as requested by Ref B
- re-wrote Sec 4 to (i) clearly state its purpose in the introductory paragraph, (ii) visually simplify the text layout thru bullet points, (iii) relegated extra info to a footnote, (iv) added a new Fig as Fig 7 d upon request by Ref B and a corresponding discussion in the text, (v) formulated a clear take-home message from the section
- moved Fig 9 to Sec 5.2 as requested by Ref B
- re-organized Sec 6
- added Fig 14 to Sec 7.2 to provide a visualization to clarify the text
- added new Sec C3 in appendix
- added references as suggested by the referees, and due to private communication with colleagues working in the field
- answered to all critique points raised by the referees

---

## Editorial Decision

published